# A chronic signaling TGFb zebrafish reporter identifies immune response in melanoma

Haley R Noonan[1,2,3,4†], Alexandra M Thornock[1,2,3,4†], Julia Barbano[1], Michael E Xifaras[1,2,3,5], Chloe S Baron[1,2,3], Song Yang[1,2,3], Katherine Koczirka[1], Alicia M McConnell[1,2,3], Leonard I Zon[1,2,3]*

[1]Stem Cell Program and Division of Hematology/Oncology, Boston Children's Hospital and Dana Farber Cancer Institute, Howard Hughes Medical Institute, Boston, United States; [2]Stem Cell and Regenerative Biology Department, Harvard University, Cambridge, United States; [3]Harvard Medical School, Boston, United States; [4]Biological and Biomedical Sciences Program, Harvard Medical School, Boston, United States; [5]Immunology Program, Harvard Medical School, Boston, United States

**Abstract** Developmental signaling pathways associated with growth factors such as TGFb are commonly dysregulated in melanoma. Here we identified a human TGFb enhancer specifically activated in melanoma cells treated with TGFB1 ligand. We generated stable transgenic zebrafish with this TGFb Induced Enhancer driving green fluorescent protein (*TIE:EGFP*). *TIE:EGFP* was not expressed in normal melanocytes or early melanomas but was expressed in spatially distinct regions of advanced melanomas. Single-cell RNA-sequencing revealed that *TIE:EGFP*+ melanoma cells down-regulated interferon response while up-regulating a novel set of chronic TGFb target genes. ChIP-sequencing demonstrated that AP-1 factor binding is required for activation of chronic TGFb response. Overexpression of *SATB2*, a chromatin remodeler associated with tumor spreading, showed activation of TGFb signaling in early melanomas. Confocal imaging and flow cytometric analysis showed that macrophages localize to *TIE:EGFP*+ regions and preferentially phagocytose *TIE:EGFP*+ melanoma cells compared to *TIE:EGFP*- melanoma cells. This work identifies a TGFb induced immune response and demonstrates the need for the development of chronic TGFb biomarkers to predict patient response to TGFb inhibitors.

*For correspondence:
zon@enders.tch.harvard.edu

†These authors contributed equally to this work

## Editor's evaluation

This is an important study that discovered a TGFb-inducible enhancer region in a human melanoma cell line that functions across vertebrates and was used to generate a zebrafish melanoma TGFb reporter model. The data is solid and provides interesting insights into TGFb signaling that is only activated in advanced melanoma. The study also shows TGFb reporter-positive melanoma cells are preferentially phagocytosed by macrophages, making this paper of interest to biologists studying melanoma and cancer immunotherapy.

## Introduction

Melanoma, arising from pigment producing melanocytes, is the deadliest form of skin cancer, with an estimated 100,640 new cases and 8,290 deaths in the United States in 2024 alone (*Siegel et al., 2024*). The most common mutation in melanoma is BRAF[V600E], which accounts for approximately

50% of melanoma cases and results in activation of the MAPK pathway promoting cell growth and survival (*Akbani et al., 2015*; *Lo and Fisher, 2014*). In addition, developmental signaling pathways are commonly dysregulated. Melanoma cells have increased expression and secretion of TGFb ligands compared to normal melanocytes, and TGFb ligand expression correlates with melanoma progression (*Van Belle et al., 1996*; *Albino et al., 1991*; *Rodeck et al., 1991*; *Rodeck et al., 1994*; *Krasagakis et al., 1998*; *Javelaud et al., 2008*; *Perrot et al., 2013*; *Rak et al., 1996*; *Cerami et al., 2012*; *Gao et al., 2013*; *Lauden et al., 2014*; *Schmid et al., 1995*). TGFb ligand binding to receptors on the cell surface results in phosphorylation and activation of SMAD2 and SMAD3 transcription factors. SMAD2 and SMAD3 translocate to the nucleus with SMAD4 to modulate gene expression (*Batlle and Massagué, 2019*). In normal melanocytes and early melanoma, TGFb acts as a tumor suppressor. However, in advanced melanoma TGFb is pro-tumorigenic as it induces growth, invasion, and metastasis (*Perrot et al., 2013*). As current targeted MAPK and immune checkpoint inhibitors often result in resistance, there is a need to study additional pathways perturbed in melanoma, such as TGFb.

Most cells in the tumor microenvironment can respond to and initiate TGFb signaling, although this often occurs in a heterogenous manner (*Derynck et al., 2021*). Generally, TGFb has an immunosuppressive effect in advanced tumors, resulting in inactivation of cytotoxic CD8$^+$ T cells, expansion of immune suppressive regulatory T cells, inhibition of dendritic cell antigen presentation, and conversion of macrophages to an anti-inflammatory and pro-angiogenic M2-like state (*Batlle and Massagué, 2019*; *Derynck et al., 2021*; *Naganuma et al., 1996*; *Ahmadzadeh and Rosenberg, 2005*; *Donkor et al., 2011*; *Fridlender et al., 2009*; *Mantovani et al., 2002*; *Standiford et al., 2011*; *Kobie et al., 2003*). TGFb can act as a chemoattractant for macrophages and monocytes to areas of inflammation. Recruitment of monocytes by TGFb results in differentiation into macrophages that attach to the extracellular matrix (ECM) or promote blood vessel leakiness allowing for tumor cell extravasation (*Arwert et al., 2018*; *Kelly et al., 2018*; *Kim et al., 2021*; *Allen et al., 1990*; *Wahl et al., 1993*). In colorectal and urothelial cancers, TGFb in the tumor microenvironment was found to mediate immune evasion such that TGFb inhibition rendered these tumors susceptible to anti-PD-1/PD-L1 immune checkpoint inhibitors (*Tauriello et al., 2018*; *Mariathasan et al., 2018*). Due to its immunosuppressive effect, several TGFb inhibitors are in clinical trials in combination with immune checkpoint inhibitors (*Lan et al., 2018*; *Paz-Ares et al., 2020*; *Strauss et al., 2018*; *Martin et al., 2020*; *Welsh et al., 2021*).

Here, we visualized TGFb response across zebrafish melanoma tumorigenesis. A human melanoma enhancer, induced upon TGFB1 signaling was identified, and stable transgenic zebrafish with this enhancer driving EGFP were generated (*TIE:EGFP*). This TGFb inducible enhancer reporter is expressed in spatially distinct regions in advanced zebrafish melanomas and is characterized by up-regulation of a series of novel chronic TGFb target genes involved in the extracellular matrix. Single-cell RNA-seq and confocal microscopy revealed that *TIE:EGFP*$^+$ melanoma cells down-regulate interferon response and are preferentially phagocytosed by macrophages. Overexpression of the chromatin remodeler *SATB2*, which is associated with tumor spreading, shows early activation of TGFb signaling in these melanomas, suggesting that specific melanoma genotypes may benefit from TGFb inhibition. Overall, this work demonstrates the need for biomarker development to predict response to TGFb inhibitors in advanced or aggressive melanoma subtypes.

## Results
### A TGFb enhancer reporter is inducible and specific in zebrafish

To visualize dynamic TGFb response across melanoma development, we designed a TGFb Inducible Enhancer (TIE) reporter using human melanoma cell (A375) ChIP-seq data following a 2-hr TGFB1 treatment. RNA-sequencing indicated that following a 2-hr treatment of A375 cells with 10 ng/mL human recombinant TGFB1, 223 genes were significantly up-regulated (q<0.05) including typical TGFb target genes SMAD7, JUNB, and PMEPA1, while 94 genes were down-regulated (*Figure 1—figure supplement 2A*, left). Hallmark gene set enrichment analysis (GSEA) of genes ranked by log2 fold-change (log2fc) confirmed that TGFb was the top up-regulated pathway following 2-hr treatment (q=0) (*Figure 1—figure supplement 2A*, right). This indicates that a 2-hr treatment with human recombinant TGFB1 ligand is sufficient to activate the TGFb pathway.

To identify a TGFB1 inducible enhancer region, we selected a region of chromatin that was exclusively open upon stimulation, based on H3K27ac ChIP-seq, with unique SMAD2/3 binding following

treatment (*Figure 1A*). The enhancer region under the SMAD2/3 peak was cloned upstream of a beta-globin minimal promoter and EGFP in a Tol2 vector backbone. *TIE:EGFP* was tested for inducibility in the presence of ubiquitously expressed constitutively active SMAD2 and SMAD3 (*ubi:caSMAD2/3*) by electroporation of adult zebrafish skin, with ubiquitous BFP (*ubi:BFP*) as a control for electroporation efficiency. *TIE:EGFP* reporter activity was significantly increased in the presence of *ubi:caSMAD2* and *ubi:caSMAD3*, confirming that the reporter is inducible (*Figure 1—figure supplement 1A*).

The *TIE:EGFP* reporter construct was microinjected into *Tg(mitfa:BRAF^V600E^);p53^-/-^;mitfa^-/-^* zebrafish in order to generate melanomas. *TIE:EGFP;Tg(mitfa:BRAF^V600E^);p53^-/-^;mitfa^-/-^* stable embryos have EGFP expression in the anterior brain at 24 hr post fertilization (hpf) extending along the brain and spinal cord beginning at 48 hpf (*Figure 1B*). This expression pattern is consistent with that previously found in a Smad3 zebrafish reporter, although in our hands this Smad3 reporter was not active in adults (*Casari et al., 2014*). *TIE:EGFP* embryos were treated at 24 hpf with 50 or 100 µM of SB431542, a TGFb type I receptor kinase inhibitor, and imaged at 48 hpf (*Kim et al., 2021*). Treatment with this TGFb inhibitor for 24 hr drastically reduced *TIE:EGFP* signal (*Figure 1—figure supplement 1B, C*). This indicates that the *TIE:EGFP* reporter is specific to TGFb signaling.

### *TIE:EGFP* reporter is expressed in advanced zebrafish melanomas

To visualize TGFb response across melanoma development, single cell *Tg(mitfa:BRAF^V600E^);p53^-/-^;mitfa^-/-^* embryos stably expressing *TIE:EGFP* were injected with an empty MiniCoopR vector (*MCR:MCS*) containing the *mitfa* minigene to generate melanomas. *Tyrosinase* gRNA and Cas9 protein were also injected to knock-out melanocyte pigment as well as *mitfa:mCherry* to allow for melanocyte visualization. Normal melanocytes rarely had *TIE:EGFP* expression, with the exception of occasional fin melanocytes (*Figure 1—figure supplement 1D*). *TIE:EGFP* has yet to be observed in *mitfa:mCherry* high early melanomas (*Figure 1—figure supplement 1E*). However, of the fish with melanoma formation (n=56), 55% turned on *TIE:EGFP* in advanced melanomas defined by protrusion from the body plane (*Figure 1C*). *TIE:EGFP^+^* cells often occur in clusters throughout the tumor and these cells are *TIE:EGFP^+^* for many weeks at a time. This data indicates that advanced zebrafish melanomas develop TGFb responsive zones.

### *TIE:EGFP* expressing melanoma cells down-regulate interferon signaling

To understand the transcriptional differences between *TIE:EGFP^+^* and *TIE:EGFP^-^* melanoma cells, single *TIE:EGFP^+^* and *mitfa:mCherry^+^* cells from *TIE:EGFP* expressing melanomas were processed for single cell RNA-sequencing at 23 and 42 weeks post fertilization (wpf), respectively (melanomas used are depicted in *Figure 1C* and *Figure 1—figure supplement 2B*). We used SORT-seq to sequence flow cytometry-sorted single cells and in silico linked transcriptomes to fluorescence intensities as recorded by FACS indexing (*Muraro et al., 2016*). A UMAP of the sequencing results from our combined replicates is shown (*Figure 2A*; UMAP with no batch correction in *Figure 2—figure supplement 1A*). Most cells were identified as melanoma cells expressing *mitfa* and/or *sox10*, but we identified an *mpeg1.1*-expressing *TIE:EGFP^+^* macrophage cluster as well (*Figure 2A*).

The melanoma cell population was separated into *TIE:EGFP^-^*, *TIE:EGFP^low^*, and *TIE:EGFP^high^* based on FACS intensities. We found that immediate-early TGFb target genes, identified following an acute 2 hr TGFB1 treatment in zebrafish melanoma cells, ZMEL1, were down-regulated in *TIE:EGFP^high^* cells (*Figure 2B*). However, we identified 29 genes that were up-regulated in *TIE:EGFP^high^* cells. We termed these genes, related to the extracellular matrix, 'chronic' TGFb targets (*Figure 2C*; *Katsuno et al., 2019*; *Verrecchia et al., 2001*). At the time of dissociation, these melanomas had *TIE:EGFP^+^* cells for several weeks, therefore they were likely not in an acute TGFb response phase (*Figure 1C* and *Figure 1—figure supplement 2B*). Using GSEA analysis we found that the top down-regulated pathways in *TIE:EGFP^high^* cells were interferon alpha and gamma (*Figure 2—figure supplement 1B*). This suggests that TGFb has an immune suppressive effect in these melanomas.

### AP-1 factors are required for induction of the chronic TIE reporter

Although the vast majority of literature focuses on acute TGFb signaling, our *TIE:EGFP* reporter remains on in melanoma, reading out chronic TGFb signal. We next asked what transcription factors are necessary for induction of our novel chronic TGFb reporter and therefore required for chronic

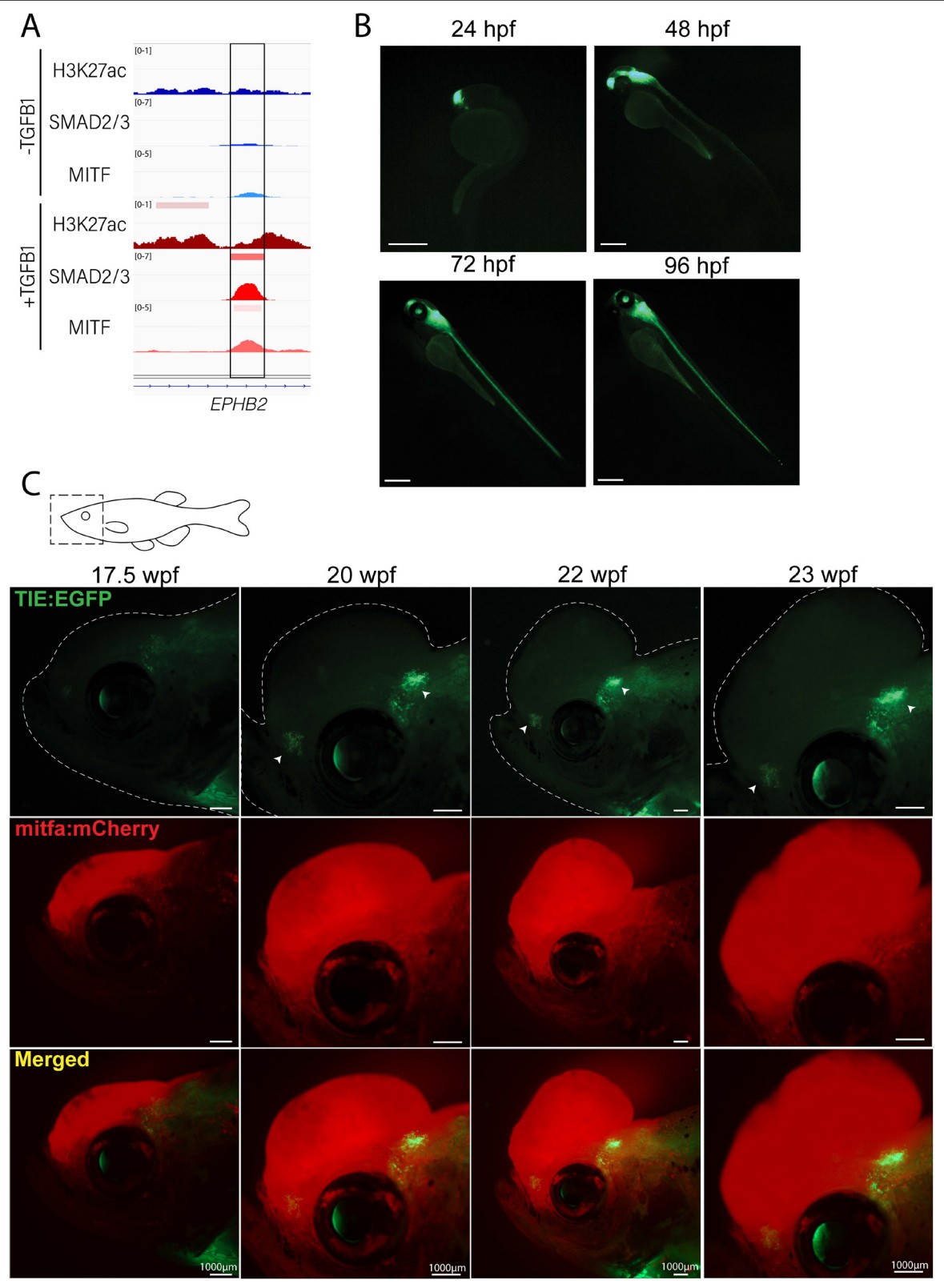

**Figure 1.** Novel *TIE:EGFP* zebrafish enhancer reporter is expressed in advanced melanomas. (**A**) TGFb-induced enhancer (TIE) used to construct *TIE:EGFP* reporter determined by H3K27ac, SMAD2/3, and MITF ChIP-seq peaks in A375s+/-TGFB1. There is unique H3K27ac and SMAD2/3 binding upon stimulus. (**B**) *TIE:EGFP* expression throughout zebrafish development. Scale bars represent 500 μm. (**C**) *TIE:EGFP* expression across

*Figure 1 continued on next page*

*Figure 1 continued*

melanomagenesis indicated by arrowheads. Representative images shown. Additional tumors shown in **Figure 1—figure supplement 3**. Illustrated fish diagram in (**C**) created with BioRender.com, and published using a CC BY-NC-ND license with permission.

The online version of this article includes the following figure supplement(s) for figure 1:

**Figure supplement 1.** *TIE:EGFP* zebrafish enhancer reporter is inducible and specific.

**Figure supplement 2.** Treatment with human recombinant TGFB1 activates the TGFb pathway in human melanoma cells.

**Figure supplement 3.** *TIE:EGFP* expression across melanomagenesis in several additional *TIE:EGFP;Tg(mitfa:BRAF^V600E^);p53^-/-^;mitfa^-/-^;MCR:MCS;mitfa:mCherry;tyr^-/-^* fish.

TGFb signaling. We performed HOMER motif analysis of ~12,000 SMAD2/3 responsive regulatory elements. These regions were identified in A375 melanoma cells stimulated with TGFB1 and were bound by SMAD2/3 upon stimulus. We found that these SMAD2/3 bound regions are highly enriched for AP-1 motifs (**Figure 2D**). ChIP-seq in A375 cells in the presence or absence of TGFB1 treatment confirmed the binding of AP-1 transcription factors JUNB and ATF3 (**Figure 2E**). ATF3 and JUNB were bound at 19% and 48% of SMAD2/3 responsive elements, respectively, before TGFB1 treatment was administered. This data indicates that AP-1 factors, in particular JUNB, may be important for SMAD binding to chromatin.

We hypothesized that AP-1 factors occupy SMAD responsive elements prior to stimulation, stabilizing open chromatin to allow the TGFb response to occur quickly upon stimulation. To test this hypothesis, we deleted AP-1 motifs in our TIE enhancer driving luciferase. We identified AP-1 motifs (highlighted in orange) in our TGFb-induced enhancer using MoLoTool and HOCOMOCO motifs (**Figure 2F**, top; **Kulakovskiy et al., 2018**). AP-1 motif #1 was the most significant (p=8.4e$^{-6}$), and AP-1 motif #2 had a lesser p-value of 8.2e$^{-4}$. We deleted either AP-1 site #1 alone, AP-1 site #2 alone, or both AP-1 sites, as well as the most significant SMAD2/3 motif (p=6.7e$^{-6}$), shown in red (**Figure 2—figure supplement 2**). In A375 cells, loss of AP-1 site #1 rendered the enhancer region unresponsive to TGFb signaling, however loss of AP-1 site #2 did not alter inducibility (**Figure 2F**, bottom). This result indicates that AP-1 site #1 is necessary for TGFb inducibility at this enhancer. Deletion of the SMAD site also destroyed TGFB1 responsiveness. This data suggests that AP-1 binding is required for responsiveness of this chronic TGFb inducible enhancer in melanoma.

## Macrophages preferentially phagocytose *TIE:EGFP* expressing melanoma cells

We identified *TIE:EGFP*$^+$ macrophages in *TIE:EGFP;Tg(mitfa:BRAF^V600E^);p53^-/-^;mitfa^-/-^;MCR:MCS* melanomas that express *mitfa* and *sox10*, suggesting recent phagocytosis of melanoma cells (**Figure 3A**). There are two potential models for why macrophages are *TIE:EGFP*$^+$: (1) macrophages express *TIE:EGFP* themselves, or (2) macrophages are engulfing *TIE:EGFP*$^+$ cells. To test these hypotheses, we crossed *TIE:EGFP;Tg(mitfa:BRAF^V600E^);p53^-/-^;mitfa^-/-^* zebrafish to a *mpeg:mCherry* reporter line that labels macrophages with mCherry (**Ellett et al., 2011**). These embryos were injected with *MCR:BRAF^V600E^*, *2x U6:p53/Tyr gRNA mitfa:Cas9,* and *mitfa:BFP* to generate pigmentless, BFP$^+$ melanomas. As the *TIE:EGFP* reporter is cytoplasmic, if *mpeg:mCherry*$^+$ macrophages are endogenously expressing *TIE:EGFP*, we would expect the entire macrophage to appear yellow via imaging. Confocal imaging of these tumors showed that the majority of a *TIE:EGFP*$^+$ region of the tumor is composed of clustered *TIE:EGFP*$^+$ macrophages. Within these regions, we observed that a subset of macrophages contain only puncta of *TIE:EGFP* signal, suggesting phagocytosis of *TIE:EGFP*$^+$ cells, while a separate subset are entirely yellow (suggesting endogenous reporter expression; **Figure 3B** and **Figure 3—figure supplement 1A**). In 10 out of 13 tumors analyzed by confocal microscopy, about 60–80% of *TIE:EGFP*$^+$ macrophages contained *TIE:EGFP* fragments, while the remaining *TIE:EGFP*$^+$ macrophages were diffusely *TIE:EGFP*$^+$. In the remaining three tumors, about 30% of *TIE:EGFP*$^+$ macrophages contained *TIE:EGFP* fragments, while the remaining were diffusely *TIE:EGFP*$^+$. Therefore, in most tumors (with rare exceptions), the majority of *TIE:EGFP*$^+$ macrophages are EGFP$^+$ because they phagocytosed a *TIE:EGFP*$^+$ cell. Indeed, we observed macrophages actively phagocytosing *TIE:EGFP*$^+$ cells (**Figure 3—figure supplement 1C**). In three out of 13 tumors imaged by confocal microscopy,

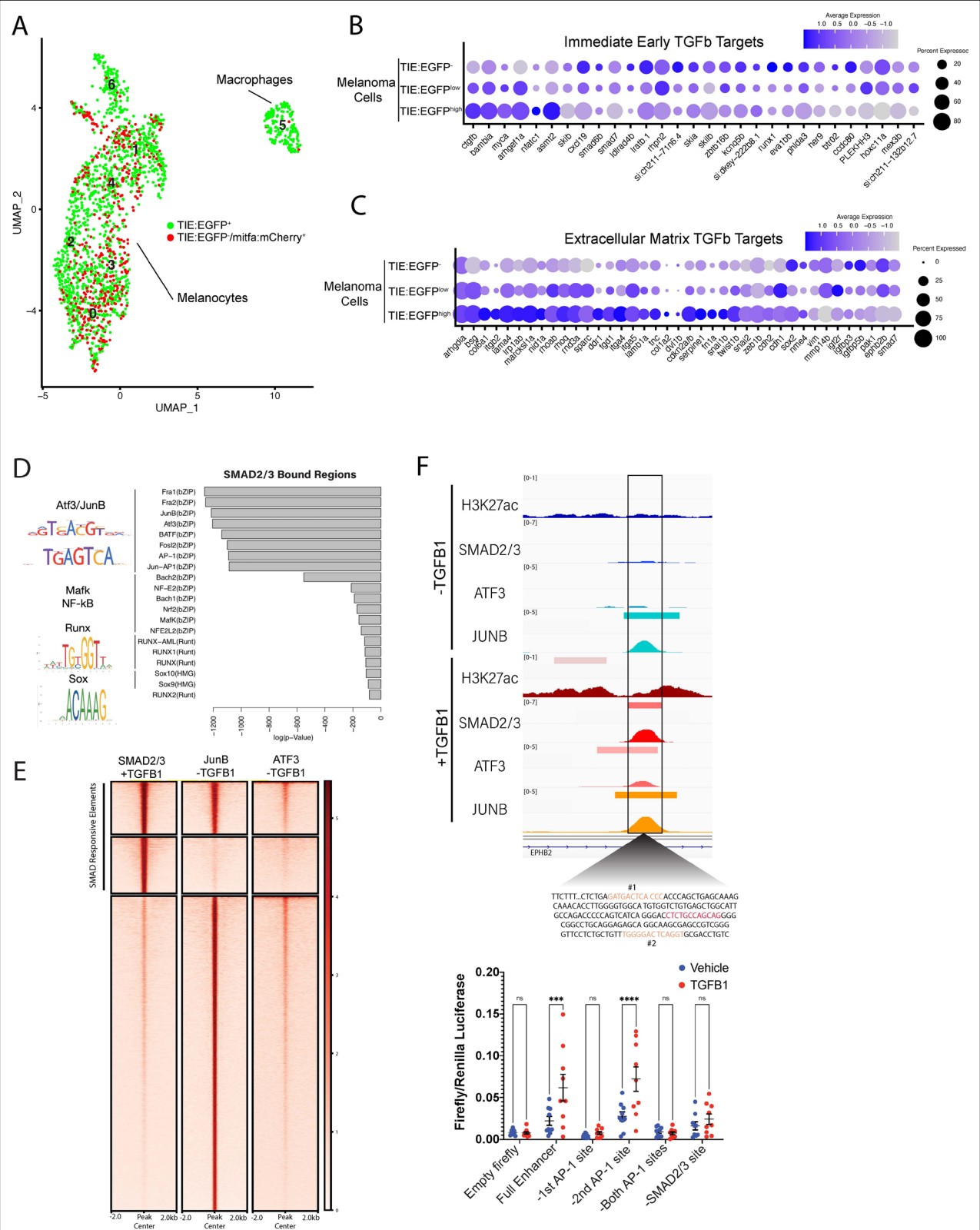

**Figure 2.** TGFb responsive melanoma cells in *MCR:MCS* tumors up-regulate chronic extracellular matrix TGFb target genes and AP-1 binding is required for TGFb responsiveness. (**A**) (Left) UMAP depicting seven cell clusters identified by SORT-seq, combined two *MCR:MCS* biological replicates. Approximately 2256 *TIE:EGFP+* cells and 752 *mitfa:mCherry+;TIE:EGFP-* cells (as a control) were sorted by flow cytometry for scRNA-seq. *TIE:EGFP+* cells were both *mitfa:mCherry+ and mitfa:mCherry-*. In analysis of the scRNA-seq data, melanoma cells were identified as being *mitfa* and *sox10* positive,

*Figure 2 continued on next page*

*Figure 2 continued*

while macrophages were identified as *mpeg1.1* and *marco* positive. (**B**) Dotplot depicting TGFb immediate-early target gene expression in *TIE:EGFP^high^*, *TIE:EGFP^low^*, and *TIE:EGFP^-^* melanoma cells. Melanoma cells can be segregated into *TIE:EGFP^high^* vs. *TIE:EGFP^low^* based on EGFP intensity during sorting. (**C**) Dotplot depicting extracellular matrix TGFb target gene expression in *TIE:EGFP^high^*, *TIE:EGFP^low^*, and *TIE:EGFP^-^* melanoma cells. (**D**) HOMER motif analysis of regulatory regions bound by SMAD2/3 upon stimulation in A375 cells. (**E**) Heatmap showing binding of JUNB and ATF3 pre-stimulus at 12,000 SMAD2/3-responsive elements in A375. (**F**) (Top) IGV plot of H3K27ac, SMAD2/3, ATF3 and JUNB ChIP-seq +/- TGFB1 stimulus at the TGFb-induced enhancer. Inset depicts sequence under SMAD2/3 ChIP-seq peak and highlights AP-1 (orange) and SMAD2/3 (red) binding sites. (Bottom) Firefly luciferase luminescence of full TIE reporter or reporter lacking AP-1 or SMAD2/3 sites. Normalized to Renilla transfection control. Experiment performed three times with three technical replicates each. A two-way multiple comparison ANOVA was used to calculate significance.

The online version of this article includes the following source data and figure supplement(s) for figure 2:

**Figure supplement 1.** TGFb responsive melanoma cells down-regulate interferon target genes.

**Figure supplement 2.** Mutated TGFb inducible enhancer sequences used for luciferase assays in *Figure 2F*.

**Figure supplement 2—source data 1.** Editable version of *Figure 2—figure supplement 2*.

we observed that in *TIE:EGFP^+^* patches, macrophages were engaged closely with a cluster of *TIE:EGFP^+^;mitfa:BFP^+^* tumor cells, likely in the process of phagocytosis (*Figure 3B* and *Figure 3—figure supplement 1B*). Together, this data reveals that a large proportion of macrophages engulf a *TIE:EGFP^+^* cell population that includes melanoma cells.

To determine if macrophages are preferentially phagocytosing *TIE:EGFP^+^* melanoma cells, three *TIE:EGFP^+^;mpeg:mCherry^+^;mitfa:BFP^+^* melanoma were excised, digested, and processed for flow cytometry analysis (*Figure 3C–D*). Viable cells were separated into *TIE:EGFP^-^* and *TIE:EGFP^+^* (*Figure 3D*, far left). In all three tumors, less than 1% of sorted cells were *TIE:EGFP^+^*, indicating this TGFb responsive population is rare. *TIE:EGFP^-^* and *TIE:EGFP^+^* (*Figure 3D*, middle) cells were plotted relative to *mpeg:mCherry* and *mitfa:BFP*. Q2 represents the total percentage of *TIE:EGFP^-^* or *TIE:EGFP^+^* cells that were melanoma cells phagocytosed by macrophages. The percentage given in Q2 is plotted for several tumors in *Figure 3D* (far right). Macrophages phagocytose a significantly higher percentage of *TIE:EGFP^+^* melanoma cells compared to *TIE:EGFP^-^* melanoma cells. This was confirmed by qPCR, which showed that *mitfa* and *sox10* expression is enriched in *TIE:EGFP^+^* macrophages compared to *TIE:EGFP^-^* macrophages (*Figure 3—figure supplement 2*). Altogether, scRNA-seq, confocal imaging and flow cytometry analyses demonstrate that macrophages preferentially phagocytose *TIE:EGFP^+^* melanoma cells compared to *TIE:EGFP^-^* melanoma cells.

## *SATB2* expression leads to early TGFb activation

The chromatin remodeler *SATB2* is amplified in 4–8% of melanoma patients and its expression correlates with patient outcome. Our lab has previously shown that *SATB2* overexpression leads to accelerated melanoma onset, produces more aggressive, metastatic tumors, and induces a TGFb signature (*Fazio et al., 2021*). *SATB2* was overexpressed under the *mitfa* promoter in the MiniCoopR backbone (*MCR:SATB2*) in *TIE:EGFP;Tg(mitfa:BRAF^V600E^);p53^-/-^;mitfa^-/-^* zebrafish. Of the fish with S*ATB2* melanomas (n=27), 68% turned on *TIE:EGFP* in tumors, and its signal was expressed throughout the tumor volume (*Figure 4A*). To understand the transcriptional differences between *TIE:EGFP^+^* and *TIE:EGFP^-^* melanoma cells, we sequenced the transcriptome of single cells from an *MCR:SATB2* expressing melanoma at 30 wpf using SORT-seq (tumor used shown in *Figure 4A*, top). We identified most cells as melanoma cells with *mitfa* and/or *sox10* expression, and again an *mpeg1.1* expressing *TIE:EGFP^+^* macrophage population (*Figure 4B* and *Figure 4—figure supplement 1A*).

To determine if there is a change in state of the macrophages that express *TIE:EGFP* (either endogenously or via phagocytosis), we subset macrophages from the *SATB2* expressing melanoma and separated macrophages that were *TIE:EGFP^+^* or *TIE:EGFP^-^* based on flow cytometry intensity. TGFb is known to convert macrophages from a pro-inflammatory M1-like to an anti-inflammatory M2-like state which can be characterized by gene expression. In the zebrafish, these include M1 markers *acod1*, *tnfa*, *csf3a/b*, and *socs3b*, as well as M2 markers *mrc1b*, *vegfaa*, *alox5ap*, and *marco* (*Viola et al., 2019*; *Wu et al., 2015*; *Jeannin et al., 2018*; *Wilson, 2014*; *Martinez and Gordon, 2014*; *Hwang et al., 2020*; *Ye et al., 2021*; *Georgoudaki et al., 2016*). We found that although *TIE:EGFP^+^* macrophages are not clearly polarized, they express a combination of M1-like and M2-like marker genes (*Figure 4—figure supplement 1B*).

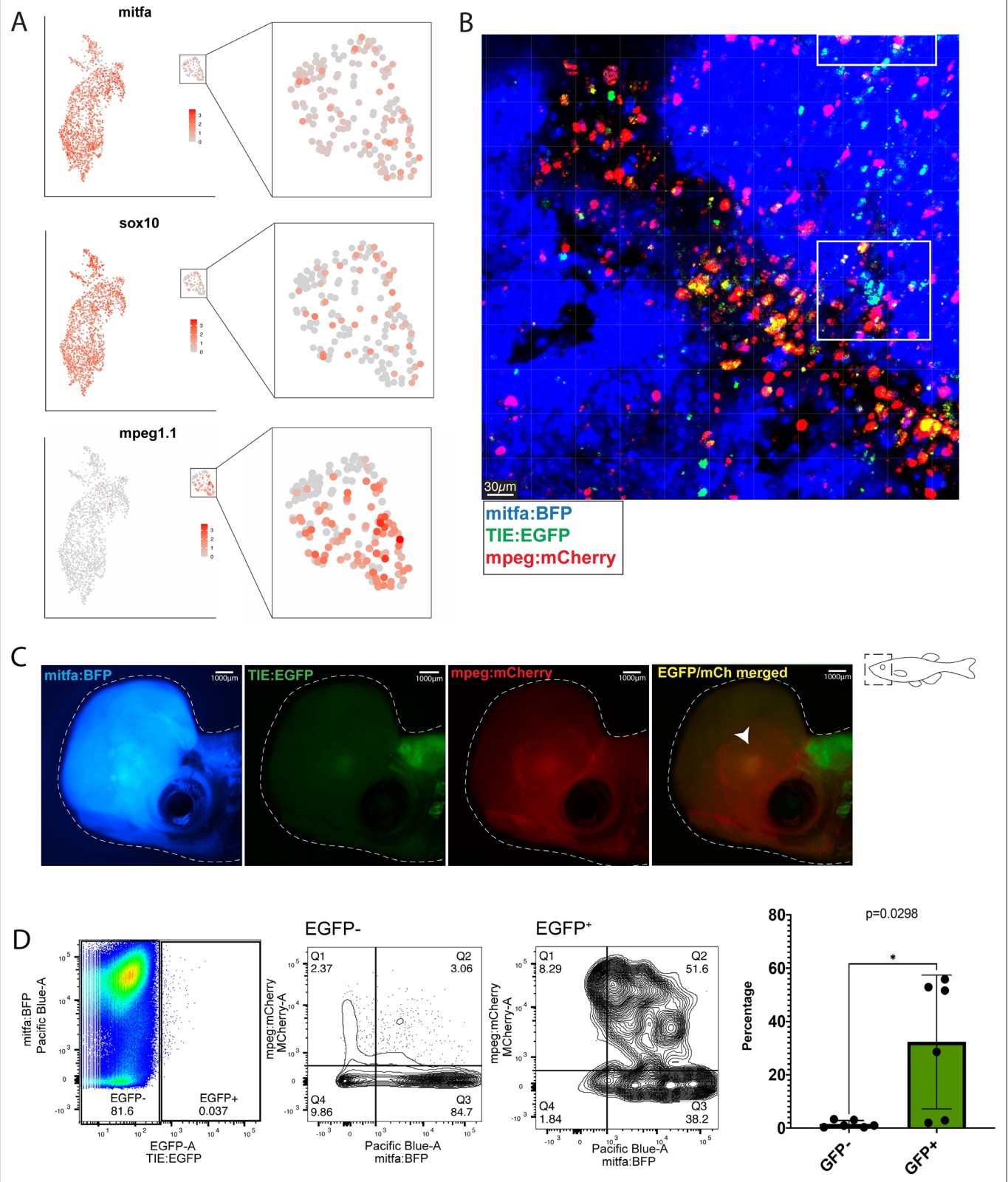

**Figure 3.** Macrophages preferentially phagocytose *TIE:EGFP⁺* cells. (**A**) UMAP depicting *mitfa, sox10,* and *mpeg1.1* expression in clusters identified by SORT-seq, combined two *MCR:MCS* melanoma replicates. Inset shows expression of these genes in the macrophage cluster. (**B**) Representative image from a zebrafish melanoma acquired on an upright confocal, n=13 fish. Additional images shown in *Figure 3—figure supplement 1*. Melanoma cells are blue, macrophages are red, and *TIE:EGFP⁺* cells are green. Yellow indicates a macrophage that has phagocytosed a TGFb responsive cell,

*Figure 3 continued on next page*

*Figure 3 continued*

which often appears as fragments within macrophages. A macrophage that expresses the *TIE:EGFP* endogenously would express EGFP throughout the entire cell, rather than in fragments. Cyan indicates a *TIE:EGFP⁺* melanoma cell. When phagocytosed by macrophages, *TIE:EGFP⁺* melanoma cells appear white, which are indicated within in the white boxes. (**C**) Representative *TIE:EGFP⁺;mpeg:mCherry⁺;mitfa:BFP⁺* melanoma used for flow analysis of macrophages. Scale bars indicate 1000 μm. The EGFP⁺ region adjacent to the tumor is endogenous *TIE:EGFP⁺* expression of the brain. (**D**) (Far left) Viable cells were separated into *TIE:EGFP⁻* and *TIE:EGFP⁺*. (Middle) FACS plots showing *TIE:EGFP⁻* and *TIE:EGFP⁺* cells relative to *mpeg:mCherry* and *mitfa:BFP*. Q1 in the *TIE:EGFP⁻* plot represents macrophages that have not phagocytosed any melanoma cells. Q2 represents macrophages that have phagocytosed *TIE:EGFP⁻* melanoma cells. Q1 in the *TIE:EGFP⁺* plot represents macrophages that have not phagocytosed melanoma cells, but rather express the *TIE:EGFP* reporter endogenously or phagocytosed a *TIE:EGFP⁺* non-melanoma cell. Q2 represents macrophages that have phagocytosed *TIE:EGFP⁺* melanoma cells. (Far right) Q2 of both plots are graphed to represent the percentage of all *TIE:EGFP⁻* or *TIE:EGFP⁺* live cells that are melanoma cells phagocytosed by macrophages. Two-tailed unpaired Welch's t-test was used to calculate significance. n=3 fish with two technical replicates each. Illustrated fish diagram in (**C**) created with BioRender.com, and published using a CC BY-NC-ND license with permission.

The online version of this article includes the following figure supplement(s) for figure 3:

**Figure supplement 1.** Macrophages phagocytose *TIE:EGFP⁺* cells.

**Figure supplement 2.** qPCR shows that *TIE:EGFP⁺* macrophages express higher levels of melanocyte markers *mitfa* and *sox10* (compared to *TIE:EGFP⁻* macrophages), validating that macrophages preferentially phagocytose TGFb responsive melanoma cells.

To understand the transcriptional differences between *TIE:EGFP⁺* and *TIE:EGFP⁻* melanoma cells, *SATB2* expressing melanoma cells were divided into *TIE:EGFP⁻*, *TIE:EGFP^low*, and *TIE:EGFP^high* cells based on flow cytometry intensity. These *MCR:SATB2* results confirmed what was observed in the *TIE:EGFP⁺ MCR:MCS* tumors. We found that in *TIE:EGFP^high* melanoma cells, chronic extracellular matrix TGFb target genes were up-regulated and the top down-regulated pathways by GSEA again were interferon alpha and gamma (*Figure 4C* and *Figure 4—figure supplement 1C*). Down-regulation of interferon suggests that the TGFb responsive melanoma cells likely evade adaptive immune responses, such as interferon-mediated antigen presentation to CD8⁺ T cells. Finally, 40% of the *MCR:SATB2* fish with melanomas (n=27) had *TIE:EGFP* expressed in early melanomas. These lesions are *mitfa:mCherry* high, indicating a proliferation of melanoma cells, but do not yet extend off the body plane. Meanwhile, 0% of *MCR:MCS* fish with melanomas (n=56) had *TIE:EGFP⁺* early melanomas (*Figure 4D*). This shows that overexpression of *SATB2* leads to acceleration of TGFb response in melanoma and suggests patients with this amplification may be more susceptible to TGFb inhibitors.

## Discussion

Here, we identified an inducible and specific TGFb enhancer reporter and visualized TGFb response throughout melanomagenesis in zebrafish. *TIE:EGFP* is off in early melanoma but is expressed in most advanced melanomas, and remains on, reading out chronic TGFb signaling. These TGFb responsive melanoma cells down-regulate interferon, up-regulate a series of novel chronic TGFb target genes, and are preferentially phagocytosed by macrophages. This work identifies a TGFb induced immune response and novel biomarkers of chronic TGFb signaling in melanoma.

We have shown that our reporter, generated from human ChIP-seq data, is able to read out chronic TGFb signaling. Most literature on developmental signaling focuses on acute signaling with stimulus for several hours in vitro. There are multiple explanations for why *TIE:EGFP* is a chronic TGFb reporter and once turned on, remains activated. Once the *TIE:EGFP* reporter is activated, it may remain active because chromatin is held in an open state by epigenetic modifications or AP-1 factors. We showed that AP-1 factors are necessary for TGFb induction of our TIE reporter and luciferase assays indicated that loss of AP-1 motifs eliminates basal levels of reporter transcription. AP-1 factors may be holding TIE chromatin open, therefore potentiating its ability to drive downstream reporter expression. Simultaneously, AP-1 factors likely hold chromatin open for other TGFb target genes to potentiate the TGFb response. This may allow for a rapid signaling response upon TGFb activation and suggests that AP-1 inhibitors could disrupt TGFb induction.

The *TIE:EGFP* scRNA-seq data indicates that melanoma cells responding to TGFb for several weeks see down-regulation of acute TGFb target genes. They exhibit up-regulation of 29 chronic

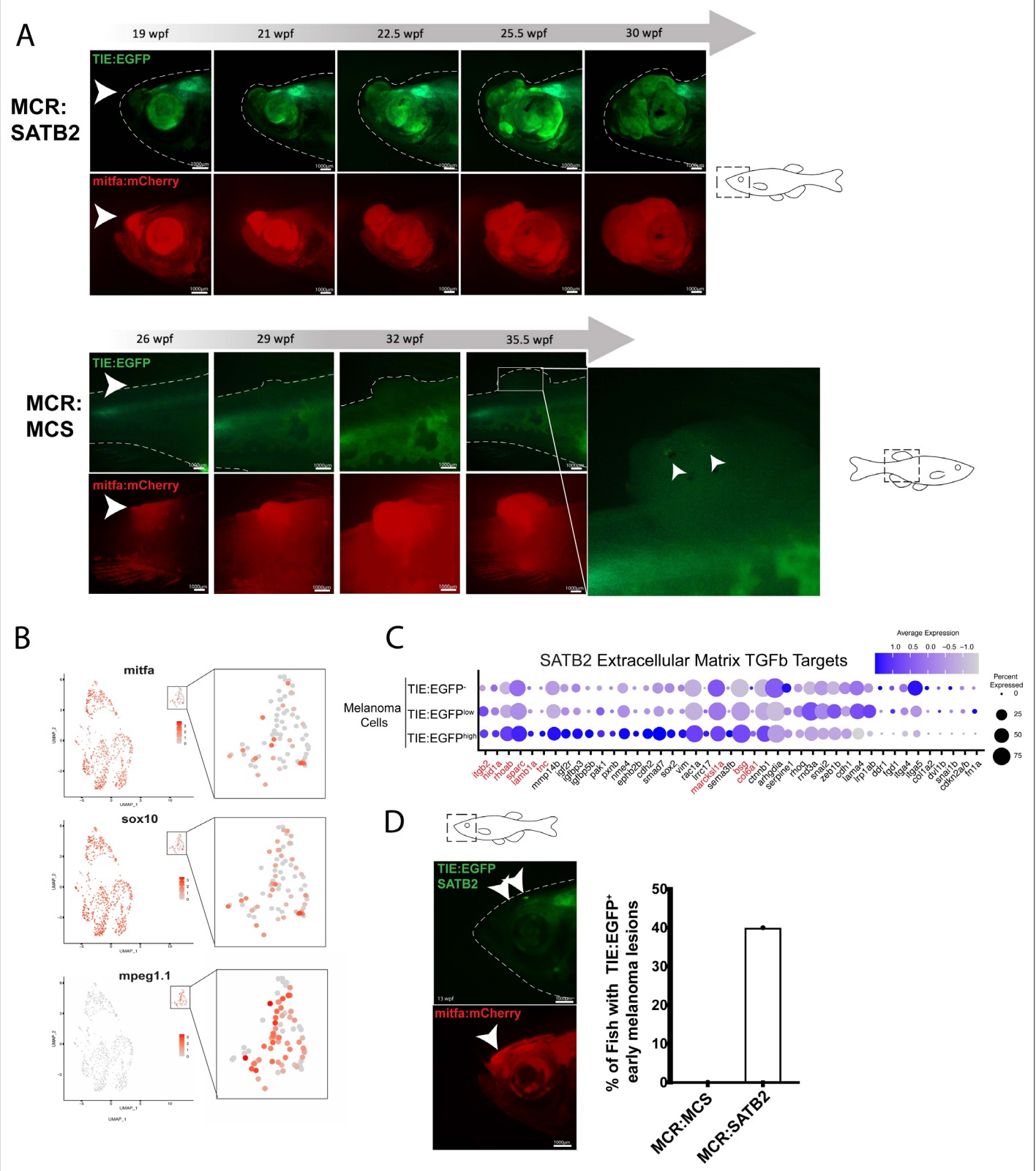

**Figure 4.** *SATB2* expressing melanomas exhibit *TIE:EGFP* expression in early initiating lesions. (**A**) (Top) Development of a representative tumor overexpressing *MCR:SATB2* in *TIE:EGFP;Tg(mitfa:BRAF^V600E^);p53^-/-^;mitfa^-/-^* zebrafish. Arrowhead indicates *TIE:EGFP*^+^ early melanoma before tumor formation. This EGFP^+^ region is separate from the endogenous *TIE:EGFP*^+^ expression of the brain. (Bottom) Development of a control *MCR:MCS* tumor in *TIE:EGFP;Tg(mitfa:BRAF^V600E^);p53^-/-^;mitfa^-/-^* zebrafish, for reference. (**B**) UMAP depicting *mitfa, sox10,* and *mpeg1.1* expression in clusters identified by

*Figure 4 continued on next page*

*Figure 4 continued*

SORT-seq of *SATB2* expressing tumor in (**A**). Inset shows expression of these genes in the macrophage cluster. (**C**) Dotplot depicting extracellular matrix TGFb target gene expression in *TIE:EGFP^high^*, *TIE:EGFP^low^*, and *TIE:EGFP^-^ SATB2* expressing melanoma cells. Genes shown in red are genes that are also upregulated in control *MCR:MCS* tumors. (**D**) Early initiating melanoma (arrowhead) overexpressing *MCR:SATB2* in *TIE:EGFP;Tg(mitfa:BRAF^V600E^);p53^-/-^;mitfa^-/-^* zebrafish. Representative image chosen. 40% of *MCR:SATB2* early melanomas (n=27) express *TIE:EGFP*, compared to 0% of *MCR:MCS* early melanomas (n=56) (quantification on right). Illustrated fish diagrams in (**A, D**) created with BioRender.com, and published using a CC BY-NC-ND license with permission.

The online version of this article includes the following figure supplement(s) for figure 4:

**Figure supplement 1.** Single-cell RNA-seq identifies *TIE:EGFP^+^* macrophages and melanoma cells in an *MCR:SATB2* tumor.

TGFb target genes that are more representative of the downstream phenotypes imposed by TGFb signal, such as extracellular matrix genes. Based on the literature, these targets degrade extracellular matrix, promoting migration (*Verrecchia et al., 2001*). Some of the chronic TGFb targets were identified in one of the few reports of long-term TGFb signaling in human mammary epithelial cells for 12 or 24 days (*Figures 2C and 4C*). Prolonged TGFb treatment was found to stabilize EMT, stem cell state, and drug resistance in breast cancer cells (*Katsuno et al., 2019*). According to TCGA data in cBioPortal, a subset of melanoma patients exhibit up-regulation of these genes (*Cerami et al., 2012*; *Gao et al., 2013*). In the future, this signature could be used as biomarkers to identify patients with chronic TGFb signaling and these patients may benefit from treatment with TGFb inhibitors.

Overexpression of *SATB2* resulted in *TIE:EGFP* expression in early melanoma lesions, which was not observed in controls. Cancer cells are thought to circumvent the tumor suppressive effects of TGFb via mutations or epigenetic modifications (*Derynck et al., 2021*; *Kim et al., 2021*; *Massagué et al., 2000*; *Zhang et al., 2017*; *Seoane and Gomis, 2017*; *Suriyamurthy et al., 2019*; *Miyazawa and Miyazono, 2017*; *Tang et al., 2018*). It is possible that melanomas up-regulating the epigenetic regulator *SATB2* may overcome the tumor-suppressive effects of TGFb signaling earlier in tumor development, leading to more aggressive and invasive melanomas (*Fazio et al., 2021*). Investigating the activation of TGFb signaling in the context of different patient mutations may provide insight into who may benefit from TGFb inhibitors, allowing for the identification of biomarkers of TGFb inhibitor response. It is expected that aggressive melanoma subtypes like *SATB2*, which activate TGFb very early in melanoma development, would be most responsive to early intervention with TGFb inhibitors.

*TIE:EGFP* expression often occurs in patches throughout the tumor. In the tumor pictured in *Figure 3C*, there is general co-localization of *mpeg:mCherry* and *TIE:EGFP* signal indicating that macrophages may cluster in TGFb positive regions of the tumor. Using confocal imaging, we confirmed that *TIE:EGFP* signal is most often found in macrophages clustered in these zones. The *TIE:EGFP* reporter appears in some macrophages as a diffuse, cytoplasmic signal indicative of endogenous TGFb signaling. However in the majority of EGFP^+^ macrophages, the reporter appears as fragments (sometimes even in endosome-like structures), indicating that many macrophages are *TIE:EGFP^+^* because they phagocytosed a *TIE:EGFP^+^* cell. We occasionally captured through confocal imaging macrophages in contact with *TIE:EGFP^+^* melanoma cells, suggesting that macrophages phagocytose these tumor cell populations (*Figure 3—figure supplement 1*). Indeed, our scRNA-seq data identified *TIE:EGFP^+^* macrophage subclusters that express *mitfa* and/or *sox10*, providing additional evidence for phagocytosis of melanoma cells. Using flow cytometry analysis, we showed that macrophages preferentially phagocytose *TIE:EGFP^+^* melanoma cells. The regionality of TGFb response could occur due to gradients in the TGFb morphogen, possibly induced by TGFb signaling within tumor cells in particular regions of the tumor. As mentioned above, TGFb can act as a chemoattractant for macrophages and induces an anti-inflammatory M2-like state. Once macrophages arrive to the TGFb region, they may phagocytose *TIE:EGFP^+^* tumor cells, thus explaining the presence of EGFP^+^ fragments within macrophages. This data suggests that local TGFb signaling within the tumor may influence macrophage localization and phagocytosis, likely leading to changes in their behavior.

One factor we identified in our single cell RNA-seq data that may mediate this interaction is *serpine1*. *Serpine1* mRNA was differentially upregulated in *TIE:EGFP^+^* melanoma cells, and its receptor, *lrp1ab* was expressed on *TIE:EGFP^+^* macrophages (data not shown). SERPINE1, which encodes plasminogen activator inhibitor-1 (PAI-1), has been shown to promote cancer cell invasiveness and macrophage

recruitment in an esophageal squamous cell carcinoma model (*Sakamoto et al., 2021*). *TIE:EGFP^high* macrophages differentially express *mmp14b* and the cysteine protease *legumain*, both of which have been described to promote TGFb bioavailability (*Bai et al., 2019*; *Sounni et al., 2010*; *Robertson and Rifkin, 2016*). Together, these preliminary transcriptional data may indicate a mechanism by which macrophages are recruited to *TIE:EGFP⁺* regions by PAI-1 gradients. Recruited macrophages may phagocytose *TIE:EGFP⁺* tumor cells and then go on to amplify the pool of active, bioavailable TGFb through the activity of enzymes such as mmp14b and lgmn. Such a model would explain the local clustering of macrophages in TGFb-responding regions of the tumor, where a subset phagocytoses *TIE:EGFP⁺* melanoma and non-melanoma cells and others experience endogenous TGFb signaling (likely from local paracrine signaling).

Single cell RNA-seq data from the *MCR:SATB2* melanoma suggests that *TIE:EGFP⁺* macrophages (which are either TGFb responsive themselves or had phagocytosed a TGFb responding cell) are not clearly polarized, expressing transcriptional markers of both M1 and M2 states. This may indicate that a transition is occurring from the M1-like to the M2-like phenotype. In our model, macrophages are attracted to TGFb expressing regions of the melanoma, and while phagocytosing in the vicinity of TGFb cytokines, begin to transition to an M2 state. M2 macrophages are known to be anti-inflammatory, immunosuppressive, and pro-angiogenic. Preliminary evidence comparing the transcriptional signatures of macrophages in the *SATB2* expressing tumor indicates that *TIE:EGFP⁺* macrophages have higher expression of cholesterol and fatty acid genes (i.e. *abca1a, npc2*) as well as apoptotic genes (i.e. *casp3b, caspa*) compared to *TIE:EGFP⁻* macrophages. This indicates that intrinsic TGFb signaling or phagocytosis of *TIE:EGFP⁺* cells may induce a stress phenotype within macrophages, resulting in death. In this case, death of macrophages over time would allow the tumor cells to evade phagocytosis. This, in conjunction with down-regulation of interferon by *TIE:EGFP⁺* melanoma cells (as seen in *MCR:MCS* and *MCR:SATB2* tumors), may lead to immune inactivation within the melanoma. Moreover, the interferon target gene, *β–2-microglobulin (b2m)* is one of the top down-regulated genes in *TIE:EGFP* expressing melanoma cells (*Figure 2—figure supplement 1B*). Loss of B2M, part of the MHC Class I molecules, is a known mechanism of immunotherapy resistance (*Alavi et al., 2018*). Together, this data suggests that activation of TGFb in melanoma is immunosuppressive, potentially by way of interferon modulation and effects on macrophage behavior and supports the need for more work on combination TGFb and immune checkpoint inhibitors.

## Methods

### Plasmids

*Tgfb-Induced-Enhancer:Beta-globin-minimal-promoter-EGFP:pA pDestTol2pA2* was cloned by PCR amplifying the enhancer (chr1:22747452–22747734) in *Figure 1A* using A375 human melanoma gDNA (Forward primer: TTCTTTGTCATCCTGGTAGAGCAAATCGAG, Reverse primer: GACAGGTC GCACCTGAGTCC) (Advantage 2 PCR Kit, Clontech #639207, Kyoto, Japan). PCR product was gel purified (Qiagen #28604, Hilden, Germany) and cloned into a pENTR-5'TOPO vector (Invitrogen #45–0711, Waltham, MA, USA). This 5 'entry vector was Gateway recombined upstream of a mouse *beta-globin-minimal-promoter-EGFP* middle entry vector, a 3 'entry polyA, and the pDestTol2pA2 backbone (ThermoFisher #12538120, Waltham, MA, USA). *Mitfa:mCherry* was Gateway recombined using the *mitfa* zebrafish promoter and a *mCherry* middle entry plasmid (ThermoFisher #12538120). *Ubi:caSMAD2* was cloned by PCR amplifying human *SMAD2* pDONR221 (DNASU HsCD00045549, Tempe, AZ, USA) using the following primers to insert constitutively active mutations (Forward: CACC ATGTCGTCCATCT TGCCATTCACGCCGCC;Reverse: CTATTCCATTTCTGAGCAACGCACTGAA GGGGATC) (Sigma 11732641001, St. Louis, MO, USA), gel purifying (Qiagen #28604) and inserting into a pENTR/D-TOPO vector (Invitrogen #45–0218). This middle entry vector was Gateway recombined using a 5' entry zebrafish ubiquitious promoter (Addgene #2732, Watertown, MA, USA), a 3' entry pA vector, and a pDestTol2pA2 backbone (ThermoFisher #12538120). *Ubi:caSMAD3* was cloned by PCR amplifying human constitutively active *SMAD3* from *Smad3 pCMV-SPORT6* (Harvard Plasmid Repository #HSCD00339271, Boston, MA, USA) (Sigma 11732641001), gel purifying (Qiagen #28604) and inserting into a pENTR/D- TOPO vector (Invitrogen #45–0218). This middle entry vector was Gateway recombined using a 5' entry zebrafish ubiquitious promoter, a 3' entry pA vector, and a pDestTol2pA2 backbone (ThermoFisher #12538120). *Ubi:BFP* was cloned via Gateway recombination

using a 5' entry zebrafish ubiquitious promoter, a 3' entry pA vector, and a pDestTol2pA2 backbone (ThermoFisher #12538120). *Mitfa:BFP* was cloned via Gateway recombination using a 5' entry zebrafish *mitfa* promoter, a 3' entry pA vector, and a pDestTol2pA2 backbone (ThermoFisher #12538120). *MCR:-SATB2* and *MCR:MCS* are from *Fazio et al., 2021*. *MCR:BRAF^V600E* (Addgene #118846) and *2xU6:p53/tyr gRNA mitfa:Cas9* (Addgene #118844) are published (*tyr* gRNA: GGACTGGAGGACTTCTGGGG; *p53* gRNA = GGTGGGAGAGTGGATGG CTG) (*Ablain et al., 2018*).

### Reporter line creation

*TIE:EGFP* reporter fish were created by injecting *TIE:Beta-globin-minimal-promoter- EGFP:pA pDest-Tol2pA2* at 6 ng/µL into single cell *Tg(mitfa:BRAF^V600E);p53^-/-;mitfa^-/-* zebrafish embryos (referred to as *TIE:EGFP;Tg(mitfa:BRAF^V600E);p53^-/-;mitfa^-/-*) (*Patton et al., 2005*). F0s were screened at 5 dpf for EGFP. At approximately 2 months post fertilization, F0s were crossed to *Tg(mitfa:BRAF^V600E);p53^-/-;mitfa^-/-* fish and F1s were screened as embryos for EGFP. EGFP+ fish were raised to adulthood.

### Electroporation

Electroporation protocols were adapted from *Callahan et al., 2018*. *Tg(mitfa:BRAF^V600E);p53^-/-;mitfa^-/-* zebrafish were anesthetized using 4% MS-222 (Pentair, TRS1, Minneapolis, MN, USA). Zebrafish were subcutaneously injected with 2 µL mix using a Hamilton syringe (Hamilton #80300, Reno, NV, USA) anterior to the dorsal fin. The following amounts of vectors were injected into each fish: 200 ng *TIE:Beta-globin-minimal-promoter-EGFP:pA pDestTol2pA2*, 400 ng *ubi:caSMAD2*, 400 ng *ubi:-caSMAD3*, 666 ng *ubi:BFP*, and 266 ng *Tol2 pCS2FA-transposase* (*Kwan et al., 2007*). DNA was prepped using Qiagen HiSpeed Plasmid Maxi Kit (QIAGEN #12662). Zebrafish were then electroporated with a BTX ECM 830 machine (BTX #45–0662 Holliston, MA, USA) using the following parameters: LV Mode; 45 V; 60ms pulse length; 5 pulses, 1 second interval. The cathode paddle was placed on the side of the fish that was injected. Electroporated fish were imaged approximately one week post electroporation using a Nikon SMZ18 Stereomicroscope (Nikon, Tokyo, Japan). To quantify TIE activity, we imaged each electroporated fish, quantified GFP intensity, and divided by the area of BFP to account for variation in electroporation efficiency.

### Stereomicroscope and confocal imaging

For stereoscope imaging, zebrafish were anesthetized using 4% MS-222 (Pentair, TRS1) and imaged using a Nikon SMZ18 Stereomicroscope using consistent imaging parameters. For confocal imaging, tumors represented in *Figure 3B* and *Figure 3—figure supplement 1C* tumors were imaged using Nikon C2si Laser Scanning confocal using a 10 X objective at 1 X magnification. Z-stacks were aligned, and images minimally processed using Imaris (RRID:SCR_007370) or ImageJ software (RRID:SCR_003070). *TIE:EGFP+* tumors depicted in *Figure 3—figure supplement 1A*, B were imaged using Zeiss LSM 980 NLO Multi-photon microscope using confocal detection at 15 X (*Figure 3—figure supplement 1A*, left), 45 X (*Figure 3—figure supplement 1A*, right), 20 X (*Figure 3—figure supplement 1B*). Z-stacks were aligned, and images were minimally processed using Imaris.

### Inhibitor treatment

*TIE:EGFP;Tg(mitfa:BRAF^V600E);p53^-/-;mitfa^-/-* F1s were crossed and F2 embryos were screened for EGFP, dechorionated at 24 hpf, and placed in a 24 well plate. Embryos were treated with either E3 zebrafish water, DMSO vehicle control (Sigma #D2650), 50 µM or 100 µM SB 431542 hydrate (Sigma # S4317-5MG). Twenty-four hours post treatment embryos were imaged using a Nikon SMZ18 Stereomicroscope (1 x objective; 3 x zoom; 7ms white light and 300ms GFP exposure used for all conditions).

### In situ hybridization

SB-treated 48 hpf embryos (*TIE:EGFP*) were fixed with 4% PFA overnight at 4°C. The embryos were then dehydrated by washes of 1:2 MeOH:PBST, then 2:1 MeOH:PBST, then 100% MeOH, and stored in 100% MeOH. Embryos were re-hydrated using washes of 2:1 MeOH:PBT, then 1:2 MeOH:PBT, then 100% PBT. Samples were then digested with Proteinase K (Sigma Aldrich, #03115828001) at 10 µg/mL and washed with PBT. Samples were then re-fixed in 4% PFA, followed by several PBT washes. Samples were incubated in 'PreHyb' solution (50% formamide, 5 X SSC, 0.1% Tween 20, citric acid to pH6, 50 µg/mL heparin, 500 µg/mL tRNA) for 4.5 hr at 70 °C. They were then

incubated in 'ProbeHyb' solution (same as PreHyb with addition of GFP1 probe at final concentration of 200 ng/500 µL) overnight at 70 °C. Samples were then consecutively washed in 'WashHyb' solution (50% formamide, 5 X SSC, 0.1% Tween 20, citric acid to pH6) at concentrations of 75%, then 50%, then 25% (in 2 X SSC) at 70 °C. Samples were then washed in 2 X SSC, then 0.2 X SSC (75%, then 50%, then 25%) in PBT. They were then incubated in blocking solution (PBT, 2% sheep serum, 2 mg/mL BSA) for 4 hr, then incubated in antibody solution (1:4000 of anti-DIG-AP (Sigma Aldrich, #11093274910) in blocking solution) overnight at 4 °C. Samples then underwent multiple PBT washes, then washing with staining wash solution (100 mM Tris HCL pH9.5, 50 mM MgCl2, 100 mM NaCl, 0.1% Tween 20). Samples were then stained (NBT 50 mg/mL, BCIP 50 mg/mL) for 30 min, then washed with PBT. Samples were post-fixed using 4% PFA at 4 °C overnight. Samples were then cleared using a series of MeOH washes: 1:2 MeOH:PBT, 2:1 MeOH:PBT, 100% MeOH. Samples were then washed with a series of glycerol washes: 30% glycerol (in PBT), 50% glycerol, 70% glycerol. They were then imaged in 70% glycerol using a stereomicroscope. Intensity of probe staining was analyzed, and embryos were binned into categories representing 'low', 'medium', or 'high', GFP intensity.

## Melanoma generation

Melanocyte development is conserved between zebrafish and mammals (*Mort et al., 2015*). The master regulator of the melanocyte lineage, MITF, is conserved in zebrafish (*mitfa*) and required for melanocyte development (*Lister et al., 1999*). Expression of *mitfa:BRAF^V600E^* together with a homozygous *p53* missense mutation, leads to development of zebrafish melanomas (*Patton et al., 2005*). A *mitfa^−/−^* mutation in *Tg(mitfa:BRAF^V600E^);p53^−/−^* fish prevents melanocyte development and spontaneous melanoma formation. Melanocyte development can then be mosaically rescued via injection of MiniCoopR (MCR), a transposon-based vector that contains a *mitfa* minigene, often alongside a candidate oncogene driven by the *mitfa* promoter (*Ceol et al., 2011*). In the absence of a candidate oncogene, the *mitfa* promoter is followed by an empty multiple cloning site (*MCR:MCS*), which is sufficient to generate melanomas in a genetic background already containing the *BRAF^V600E^* and *p53^-/-^* mutations. *Tyrosinase*, an enzyme required for melanocyte pigmentation, is also knocked out in our model to generate unpigmented melanomas, allowing for easy imaging of fluorescent reporters. *TIE:EGFP;Tg(mitfa:BRAF^V600E^);p53^-/-^* tumors were generated by injecting single cell *TIE:EGFP;Tg(mitfa:BRAF^V600E^);p53^-/-^;mitfa^-/-^* zebrafish embryos with MiniCoopR Multiple Cloning Site (*MCR:MCS*) at 20 ng/µL, *mitfa:mCherry* at 10 ng/µL, *tyr* gRNA (GGACTGGAGGACTTCTGGGG) at 10 ng/µL, Cas9 protein (PNA Bio CP02) at 50 ng/µL, and *Tol2* mRNA at 20 ng/µL (*White et al., 2008*). Experiments involving the *mpeg:mCherry* reporter line required us to cross the *TIE:EGFP;Tg(mitfa:BRAF^V600E^);p53^-/-^* zebrafish line with a Casper (*roy^-/-^;mitfa^-/-^*) *mpeg:mCherry* line, so progeny would be heterozygous for the *BRAF^V600E^* and *p53^-/-^* mutations. We therefore generated *TIE:EGFP;mpeg:mCherry;Tg(mitfa:BRAF^V600E^);p53^-/-^* tumors by injecting into the one-cell stage *MCR:BRAF^V600E^* at 10 ng/µL, *2x U6:p53/Tyr gRNA mitfa:Cas9* at 10 ng/µL, *mitfa:BFP* at 10 ng/µL, *tyr* gRNA at 10 ng/µL, Cas9 protein at 50 ng/µL, and *Tol2* mRNA at 20 ng/µL. *Tyrosinase* gRNA was created by annealing a *tyr* oligo template (CCTCCATACGATTTAGGTGACACTATAGGACTGGAGGACTTCTGGGGGTTTTAGAG CTAGAAAT AGCAAG) to a constant oligonucleotide (AAAAGCACCGACTCGGTGCCACTT TTTCAAGTTGAT AACGGACTAGCCTTATTTTAACTTGCTATTTCTAGCTCTAAAAC). The annealed oligo was filled in using T4 DNA polymerase (New England BioLabs, M0203S, Ipswich, MA, USA), PCR amplified, gel purified, transcribed (MEGAscript T7/SP6 Thermo Fisher Scientific, AM1333), and cleaned up (Zymo Research, R2051, Irvine, CA, USA). Embryos were grown to adulthood and were monitored for tumor development around 8–12 wpf.

## Zebrafish work

All zebrafish (*Danio rerio*) work was performed in accordance with the Guide for the Care and Use of Laboratory Animals of the National Institutes of Health. Animal research protocols were approved by the Institutional Animal Care and Use Committee of Boston Children's Hospital, Protocol #20-10-4254R. All zebrafish work operated according to the guidelines of the Institutional Animal Care and Use Committee of Boston Children's Hospital.

## SORT-seq

Tumors were excised and dissociated for 30 min with occasional chopping using 0.075 mg/mL Liberase (Sigma #5401119001) in DMEM (Gibco #11965–092, Waltham, MA, USA) with 1% Penstrep (Corning #30–002 CI, Corning, NY, USA). Casper zebrafish skin was used to exclude autofluorescence when setting gates (*White et al., 2008*). A *mitfa:mCherry⁺/EGFP⁻* tumor from a *TIE:EGFP;mpeg:mCherry;Tg(mitfa:BRAF^{V600E});p53^{-/-}* fish or *mitfa:mCherry;crestin:EGFP;tyr^{-/-}* zebrafish skin was used to set the gates for mCherry intensity. *Ubi:EGFP* zebrafish skin was used to set the gate for EGFP intensity. Dissociated samples were filtered through a 40 µm filter and resuspended in FACS buffer (PBS/10%FBS/1% Penstrep) before filtering through a FACS tube (Corning #352235). Single cells were sorted into 384-well-cell-capture plates containing barcoded primers from Single Cell Discoveries (https://www.scdiscoveries.com/) using a BD FACS ARIA II sorter. SYTOX was used as a live/dead marker (ThermoFisher #S34857). Library preparation and Illumina sequencing was performed by Single Cell Discoveries (*Muraro et al., 2016*). SORT-seq data are demultiplexed and aligned to zebrafish Ensembl GRCz11 annotation using scruff R packages with the following parameter (bcStart = 7, bcStop = 14, bcEdit = 1, umiStart = 1, umiStop = 6, keep = 60) (*Hao et al., 2021*, *Wang et al., 2019*). The 384 SORT-seq barcodes are downloaded from https://github.com/anna-alemany/transcriptomics/blob/master/mapandgo/bwamap/bc_celseq2.tsv ( *Alemany, 2018*). Analysis was completed in R Studio using Seurat (RRID:SCR_016341) (min.features=200; 600<nFeature RNA <10000; percent.mt <10; obj.resolution=0.2; GFPhigh>4000) (*Hao et al., 2021*; *Stuart et al., 2019*; *Butler et al., 2018*; *Satija et al., 2015*). Batch correction was performed using FindVariableFeatures and FindIntegrationAnchors, nfeatures = 15,000. Pathway analysis was conducted using Gene Set Enrichment Analysis (GSEA; RRID:SCR_003199) version 4.1.0 (*Mootha et al., 2003*; *Subramanian et al., 2005*). Zebrafish genes were converted to human using DIOPT Ortholog Finder version 8.5 (RRID:SCR_021963) and the best match was used for GSEA analysis (*Hu et al., 2011*).

## Flow analysis

Tumors were excised and dissociated for 30 min with occasional chopping using 0.075 mg/mL Liberase (Sigma #5401119001) in DMEM (Gibco #11965–092) with 1% Penstrep (Corning #30–002 CI). Casper zebrafish skin was used to exclude autofluorescent cells (*White et al., 2008*). *Mpeg:mCherry⁺* and *ubi:EGFP⁺* zebrafish skin was used to set the gates for mCherry and EGFP intensity, respectively. A *mitfa:BFP⁺* patch of skin as well as *flk:BFP⁺* skin were used to set the gates for BFP intensity. Dissociated samples were filtered through a 40 µm filter and resuspended in FACS buffer (PBS/10%FBS/1% Penstrep) before filtering through a FACS tube (Corning #352235). DRAQ7 was used as a live/dead marker (Abcam #ab109202, Cambridge, UK). Cells were analyzed using a BD FACS Aria II 5 Laser System. Two technical replicates of 1 million cells were sorted from each tumor, for a total of 2 million cells per tumor. Data was processed using FlowJo version 10.8.1 (RRID:SCR_008520). Debris, doublets, and autofluorescent cells were removed from the analysis and viable cells were separated into *TIE:EGFP⁻* and *TIE:EGFP⁺*.

## Human and zebrafish melanoma cell culture and treatment

Human melanoma A375 and zebrafish melanoma ZMEL1 cells were grown in filter sterilized DMEM (Gibco #11965–092) supplemented with 10% heat-inactivated FBS (Gemini # 900–108), 1% PenStrep (Corning #30–002 CI), and 1% Glutamine (ThermoFisher # 25030164). A375 cells were obtained from ATCC (RRID:CV-CL_0132) and grown at 37 °C, 5% $CO_2$. Cell identity was confirmed by fingerprint every 2 years and tested for mycoplasma approximately every 2–4 weeks using Lonza's second generation myco PLUS kit. ZMEL1 cells were grown at 28 °C, 5% $CO_2$ and tested for mycoplasma approximately every 2–4 weeks (*Heilmann et al., 2015*). Human recombinant TGFB1 (R&D 240-B-002, Minneapolis, MN, USA) was reconstituted at 20 µg/mL in sterile 4 mM HCl containing 1 mg/mL BSA according to manufacturer's instructions. To activate the TGFb pathway, cells were serum starved for 2 hr, then treated with 10 ng/mL human recombinant TGFB1 or 4 mM HCl containing 1 mg/mL BSA vehicle control for an additional 2 hr.

## RNA-sequencing

RNA-seq was performed in triplicate using A375 or ZMEL1 melanoma cells. RNA was collected from adherent cells using the Qiagen RNeasy Plus Mini Kit (Qiagen #74134). RNA quality was confirmed

using a Fragment Analyzer. One microgram of RNA was ribodepleted using NEBNext rRNA Depletion Kit (NEB #E6310). Ribodepleted RNA was fragmented, reverse transcribed, and library prepped (NEB #E7530, NEB #E7335). Samples were sequenced on an Illumina Hi-Seq 4000 sequencer. Quality control of RNA-Seq datasets was performed by FastQC (https://www.bioinformatics.babraham.ac.uk/projects/fastqc/) (RRID:SCR_014583) and Cutadapt (RRID:SCR_011841) to remove adaptor sequences and low quality regions (*Martin, 2011*). The high-quality reads were aligned to Ensembl build version GRCh38 of human genome or zebrafish Ensembl GRCz11 annotation (RRID:SCR_002334) using Tophat 2.0.11 (RRID:SCR_013035) without novel splicing form calls (*Trapnell et al., 2009*). Transcript abundance and differential expression were calculated with Cufflinks 2.2.1 (RRID:SCR_014597) (*Trapnell et al., 2010*). FPKM values were used to normalize and quantify each transcript; the resulting list of differentially expressed genes were filtered by log fold-change and q-value. Pathway analysis was conducted using Gene Set Enrichment Analysis (GSEA) version 4.1.0 Hallmark gene sets (*Mootha et al., 2003*; *Subramanian et al., 2005*). Deseq2 (RRID:SCR_015687) was used to create differential expression heatmaps and volcano plots. Read counts less than 10 were excluded.

## ChIP-sequencing

A375 cells were fixed directly in 15 cm plates with 11% formaldehyde and collected by scraping. Approximately 100,000,000 cells were used per condition. Cells were lysed using lysis buffers with protease inhibitors (Roche #05056489001, Basel, Switzerland) and sonicated such that fragmented chromatin was 200–300 bp long. Optimal chromatin length was confirmed by gel electrophoresis. Prior to antibody addition, 50 µL chromatin was collected for input sample. The remaining sonicated chromatin was incubated overnight at 4 °C with 10 µg antibody attached to Dynabeads (Invitrogen #10004D). Samples (including inputs) were washed with wash buffers and eluted for 6 hr at 70 °C, treated with RNaseA (Sigma #R4642) and Proteinase K (Life Technologies #AM2546, Carlsbad, CA, USA), and purified using Zymo ChIP DNA Concentrator kit (Genesee Scientific #11–379 C, San Diego, CA, USA). Libraries were end repaired (VWR #ER81050, Radnor, PA, USA), polyA tailed (Invitrogen #18252–015, NEB #M0212L), adaptor ligated (NEB #E7335), size selected using Ampure XP beads (Beckman Coulter #A63881, Brea, CA, USA, Life Technologies #12027), and PCR amplified (NEB #M0531). Libraries were run on an Illumina Hi-Seq 4000 sequencer. The following antibodies were used: H3K27ac (Abcam #4729; RRID:AB_2118291), SMAD2/3 (Abcam #202445), MITF (Sigma #HPA003259; RRID:AB_1079381), ATF3 (Abcam #207434; RRID:AB_2734728), JUNB (CST #3753, Danvers, MA, USA; RRID:AB_2130002). Using HOMER analysis (RRID:SCR_010881) we confirmed that JUNB and ATF3 binding motifs were present under their respective ChIP peaks (ATF3, p=1e$^{-4983}$) (JUNB, p=1e$^{-15697}$). All ChIP-Seq datasets were aligned to Ensembl build version GRCh38 of the human genome using Bowtie2 (version 2.2.1; RRID:SCR_016368) with the following parameters: --end-to-end, -N0, -L2086. MACS2 version 2.1.0 (RRID:SCR_013291) peak finding algorithm was used to identify regions of ChIP-Seq peaks, with a q-value threshold of enrichment of 0.05 for all datasets (*Langmead and Salzberg, 2012*; *Zhang et al., 2008*). Uropa (Universal Robust Peak Annotator) is utilized to annotate ChIP-seq peaks to neighboring genes according to Ensembl gene annotation (*Kondili et al., 2017*). The parameters are defined as proximal promoter: 500 bp upstream – 50 bp downstream of TSS; distal promoter: 2 k bp upstream – 500 bp downstream of TSS; enhancer: 100 k bp from TSS. The genome-wide transcription factor SMAD2/3, JUNB, ATF3 occupancy profile figures were generated by deeptools2 according to two computation modes (*Ramírez et al., 2016*). In the reference-point mode, a set of genomic positions (e.g. the center of ChIP peak) are used as anchor point, 2 kb upstream and downstream of these position are plotted in the profile figure. HOMER analysis was performed to confirm transcription factor binding under peaks (*Heinz et al., 2010*). The hg19 genome was used with a random set of background peaks for motif enrichment.

## Luciferase assays

Firefly luciferase reporter constructs (pGL4.24) were created by cloning the full and mutated TGFb enhancers upstream of the minimal promoter using BglII and XhoI sites (see *Figure 2—figure supplement 2* for sequences). A375 cells were plated in opaque-walled 96-well plates (Thermo Fisher #136101) and approximately 5000 cells were co-transfected with 100 ng firefly and 10 ng Renilla luciferase plasmids using Lipofectamine 3000 (Invitrogen #L3000008). After 48 hr cells were serum starved for 2 hr and treated with 10 ng/mL TGFB1 or 4 mM HCl containing 1 mg/mL BSA vehicle

control for an additional 2 hr. Firefly and Renilla luciferase were then measured using the Dual-Glo Luciferase Assay (Promega #E2920, Madison, WI, USA) according to the manufacturer's instructions. Luminescence was read on a Synergy Neo plate reader and the ratio of firefly to Renilla luminescence was calculated. Empty firefly luciferase vector was used as a negative control and Renilla luciferase was used as control for transfection efficiency. Experiments were performed in biological triplicate with three technical replicates each.

## qPCR

Two *TIE:EGFP;mpeg:mCherry;Tg(mitfa:BRAF^V600E^);p53^-/-^;mitfa:BFP* tumors were dissociated for 30 min with occasional chopping using 0.075 mg/mL Liberase (Sigma #5401119001) in DMEM/F12 (Gibco # 11580546), and the reaction was inactivated by adding a solution of 15% FBS in DMEM/F12. Dissociated samples were filtered through a 40 µm filter and resuspended in FACS buffer (PBS/5% BSA). Casper zebrafish skin was used to exclude autofluorescent cells, and *mpeg:mCherry^+^* zebrafish skin was used to set the mCherry gate. SYTOX red dead cell stain was used as a live/dead marker (Invitrogen #S34859). Cells were sorted into FACS buffer in two separate tubes: *mpeg:mCherry^+^ TIE:EGFP^+^* and *mpeg:mCherry^+^ TIE:EGFP^-^*. RNA was extracted from cells sorted by FACS using the Direct-zol RNA MicroPrep Kit (Zymo Research #2060), and reverse transcribed to cDNA using the SuperScript VILO cDNA Synthesis Kit (Invitrogen #11754–050). qPCR was performed using SYBR Green qPCR Mix (ThermoFisher # 4309155), and samples were run using the QuantStudio 6 Flex system (ThermoFisher). *Mitfa* primer sequences (FP: 5'CTACGACAGCCCAAACAAGG, RP: 5 'GCCATTGTCATGTTCGTCCA). *Sox10* primer sequences (FP: 5 'ACGCTACAGGTCAGAGT CAC, RP: ATGTTGGCCATCACGTCATG). Data was analyzed using the delta delta Ct method. Ct values were normalized to those of housekeeping gene, *b-actin*. *B-actin* primer sequences (FP: CGAGCAGGAGATGGGAACC, RP: CAACGGAAACGCTCATTGC).

## Statistics

To calculate significance of electroporation and flow analysis assays, a two-tailed unpaired t-test with Welch's correction was performed using GraphPad Prism version 9.0.2 (RRID:SCR_002798) for Mac (GraphPad Software, San Diego, California USA, https://www.graphpad.com). To calculate significance of luciferase assays, a 2-way multiple comparison ANOVA test was performed using GraphPad Prism version 9.0.2 for Mac. For the qPCR experiment in *Figure 3—figure supplement 2*, 2-tailed unpaired t-tests was performed using GraphPad Prism for Mac.

## Acknowledgements

We thank Christian Lawrence, Kara Maloney, Shane Hurley, Lauren McKay, Li- Kun Zhang, and Andrew Kowalczyk for fish care throughout this study. We also thank the Boston Children's Hospital Flow Cytometry Research Facility for assistance with FACS experiments, as well as Single Cell Discoveries for processing SORT-seq samples.

We thank the Harvard Center for Biological Imaging (RRID:SCR_018673) for infrastructure and support. The authors thank their colleagues for discussion and reading of this manuscript. Figures were created with Biorender.com. Leonard I Zon is a Howard Hughes Medical Institute Investigator. Additional funding for this work was provided by the Human Frontier Science Program (HFSP LT000494/2020 L, to C.S.B.).

# Additional information

### Competing interests

Leonard I Zon: L.I.Z. is a founder and stockholder of Fate Therapeutics, CAMP4 Therapeutics, Amagma Therapeutics, Scholar Rock, and Branch Biosciences. He is a consultant for Celularity and Cellarity. The other authors declare that no competing interests exist.

## Funding

| Funder | Grant reference number | Author |
|---|---|---|
| National Institutes of Health | NCI R01CA103846 | Haley R Noonan<br>Alexandra M Thornock<br>Julia Barbano<br>Michael Xifaras<br>Chloe S Baron<br>Song Yang<br>Katherine Koczirka<br>Alicia M McConnell<br>Leonard I Zon |
| Melanoma Research Alliance | Grant given via University of Edinburgh United Kingdom project number TBD | Alexandra M Thornock<br>Michael Xifaras<br>Chloe S Baron<br>Song Yang<br>Leonard I Zon |
| National Institutes of Health | P01CA163222 | Haley R Noonan<br>Alexandra M Thornock<br>Julia Barbano<br>Michael E Xifaras<br>Chloe S Baron<br>Song Yang<br>Katherine Koczirka<br>Alicia M McConnell<br>Leonard I Zon |

The funders had no role in study design, data collection and interpretation, or the decision to submit the work for publication.

## Author contributions

Haley R Noonan, Conceptualization, Data curation, Formal analysis, Validation, Investigation, Visualization, Methodology, Writing – original draft, Writing – review and editing; Alexandra M Thornock, Validation, Investigation, Visualization, Writing – review and editing; Julia Barbano, Data curation, Formal analysis, Validation, Investigation, Visualization, Methodology; Michael E Xifaras, Data curation, Investigation, Visualization, Writing – original draft, Writing – review and editing; Chloe S Baron, Data curation, Formal analysis, Writing – original draft, Writing – review and editing; Song Yang, Formal analysis; Katherine Koczirka, Alicia M McConnell, Data curation, Investigation; Leonard I Zon, Conceptualization, Resources, Supervision, Funding acquisition, Investigation, Methodology, Writing – original draft, Writing – review and editing

## Author ORCIDs

Haley R Noonan ⬤ http://orcid.org/0000-0002-0519-4065
Alexandra M Thornock ⬤ http://orcid.org/0000-0003-0511-4376
Leonard I Zon ⬤ https://orcid.org/0000-0003-0860-926X

## Ethics

All zebrafish (Danio rerio) work was performed in accordance with the Guide for the Care and Use of Laboratory Animals of the National Institutes of Health. Animal research protocols were approved by the Institutional Animal Care and Use Committee of Boston Children's Hospital, Protocol #20-10-4254R. All zebrafish work operated according to the guidelines of the Institutional Animal Care and Use Committee of Boston Children's Hospital.

## Decision letter and Author response

Decision letter https://doi.org/10.7554/eLife.83527.sa1
Author response https://doi.org/10.7554/eLife.83527.sa2

# Additional files

## Supplementary files
• MDAR checklist

## Data availability

Sequencing datasets can be accessed on GEO at # GSE213360. The Tgfb-Induced-Enhancer:Beta-globin-minimal-promoter-EGFP:pA pDestTol2pA2 plasmid is available on Addgene (Plasmid #220508), and the TIE:EGFP zebrafish reporter line is available upon request.

The following dataset was generated:

| Author(s) | Year | Dataset title | Dataset URL | Database and Identifier |
|---|---|---|---|---|
| Noonan HR, Thornock AM, Barbano J, Xifaras M, Baron CS, Yang S, Koczirka K, McConnell AM, Zon LI | 2024 | A chronic signaling TGFb zebrafish reporter identifies immune response in melanoma | https://www.ncbi.nlm.nih.gov/geo/query/acc.cgi?acc=GSE213360 | NCBI Gene Expression Omnibus, GSE213360 |

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
