## [Editor Report]

This is an important study that discovered a TGFb-inducible enhancer region in a human melanoma cell line that functions across vertebrates and was used to generate a zebrafish melanoma TGFb reporter model. The data is solid and provides interesting insights into TGFb signaling that is only activated in advanced melanoma. The study also shows TGFb reporter-positive melanoma cells are preferentially phagocytosed by macrophages, making this paper of interest to biologists studying melanoma and cancer immunotherapy.

---

## [Decision Letter]

**Decision letter after peer review:**

Thank you for submitting your article "A chronic signaling TGFb zebrafish reporter identifies immune response in melanoma" for consideration by *eLife*. Your article has been reviewed by 3 peer reviewers, one of whom is a member of our Board of Reviewing Editors, and the evaluation has been overseen by Paivi Ojala as the Senior Editor. The following individual involved in the review of your submission has agreed to reveal their identity: Marina C Mione (Reviewer #3).

Essential revisions (for the authors):

Although the reviewers are very positive about the manuscript, there are several areas that could be explored further. There was a consensus that the role of macrophages in particular needs more data for the model to be supported.

1. There are multiple issues with Figures 1 and Extended Data Figure 1 as seen in the reviewers' comments below. These relate to quality and quantification and should be addressed.

2. The reviewers also suggested that Figure 3 data be improved with better confocal images and time-lapse imaging.

3. In Figure 4, SATB2 data should be compared to Figures 1-3 without SATB2. Does SATB2 facilitate melanoma progression?

*Reviewer #1 (Recommendations for the authors):*

Data exists of spatially mapped transcriptomes in zebrafish melanoma:

Hunter, et al. Spatially resolved transcriptomics reveals the architecture of the tumor-microenvironment interface. Nat Commun 12, 6278 (2021).

Are the results in this study showing regions of active TGFb signaling in advanced melanoma consistent with existing data of endogenous expression patterns?

The proposed model of preferential phagocytosis of TIE:GFP+ cells vs negative cells is interesting, however, the data is still somewhat preliminary. Higher resolution imaging data and possibly time-lapse would better support the model. Could the authors provide some candidate factors produced by TIE:GFP+ cells that could trigger this response?

In the Extended Data Figure 1C-D experiments, if the mitf:mCherry is transiently injected, then the conclusions may be overstated. Line 90: "TIE:EGFP was also never expressed in mitfa:mCherry high early melanomas (Extended Data Figure 1D)". How many melanomas were observed? What level of GFP expression is required to score as positive in TIE:GFP fish? The conclusions should be reconsidered because transient F0 transgenics are inherently mosaic.

Extended data 1B images are small and difficult to evaluate. Need better images that are quantified for fluorescence intensity.

In the Figure 1C (20 and 22 wpf) panels, why is there a yellow signal in the mitfa:mCherry channel alone in the majority of the tumor but not in the merged image?

Which data does S1B refer to in this line?

Line 115: melanomas were processed for single-cell RNA-sequencing at 23 and 42 weeks post-fertilization (wpf), respectively (Figure 1C and S1B).

Could the authors describe the marco marker and its significance here?

Line: 120 melanoma cells expressing mitfa and/or sox10, but we identified a marco-expressing TIE:EGFP+.

Does the analysis of Figure S2B include macrophages or is it only the melanomas? Is there any independent validation of the GSEA results? This is regarding:

Line: 130 in TIE:EGFPhigh cells were interferon α and γ (Figure S2B).

In the following line, which genes are referred to specifically? Are these genes associated with a specific table or figure?

Line: 293 of melanoma patients exhibit up-regulation of these genes52,53.

Are there experiments that could better support the following statement:

Line: 306 Our work identified macrophage subclusters that contain low levels of mitfa and/or sox10 expression suggestive of phagocytosis of melanoma cells.

Is there a GFP/mCherry merged image to support this statement, as it is not clear from the figure?

311 Figure 3C, there is co-localization of mpeg:mCherry and TIE:EGFP signal.

The following figure legend is insufficient and needs more details (e.g. live or fixed sample? Optical section or z projection? Single frame of time-lapse?):

Line: 943 Supplement Figure 4. Example of a macrophage actively phagocytosing a TIE:EGFP+ cell in a melanoma.

*Reviewer #2 (Recommendations for the authors):*

1. Figure 1, Extended data 1 and supplemental Figure 1: better characterization of the reporter lines is important at different ages and in melanomas. Only one melanoma is shown in Figure 1C and another one in S1B, and, curiously, the pattern of TIE:EGFP expression is completely different: clusters vs homogenous expression. More pictures and quantitation of TIE:EGFP+ and TIE:EGFP- cells are required.

2. Figure 3 and results about macrophages: Figure 3A: most macrophages do not express mitfa and sox10. Figure 3B is not convincing: I do not see any white cells that would represent macrophages taken up TIE:EGFP+ melanoma cells. I can see several green macrophages that may indicate that macrophages express the TIE:EGFP construct. Additional pictures and video, if possible are needed.

3. Figure 3D: I do not understand why Q1 represent non-melanoma cells that were phagocytosed. I think these are non-phagocytosing macrophages. So, Q2 has to be plotted not Q1+Q2. Anyway, I think that allotransplant experiments of melanoma cells from TIE:EGFP fish need to be performed in mpeg1:cherry recipient, and vice versa, to demonstrate that macrophages are phagocytosing TIE:GFP+ cells rather than expressing the construct. In addition.

4. Figure 4 and SATB2 data: these data should be compared with those obtained in Figure 1-3 w/o SATB2. Does SATB2 facilitate melanoma progression? Is there any difference in the TGFB targets between control and SATB2 TIE:EGFP+ cells?

5. Lines 286-295: it is speculated that the novel chronic TGFB targets are involved in cell migration. I think it would be a nice complement to perform an allotransplant of TIE:EGFP+ and TIE:EGF- melanoma cells and study their aggressiveness.

*Reviewer #3 (Recommendations for the authors):*

1) The confocal pictures in Figure 3B are not convincing. We cannot see the edge of the cells to make sure that they are not debris. At the same time, how do we expect that melanoma cells that have been phagocytosed by macrophages and that are present in fragments (only puncta of TIE:EGFP signal are seen in macrophages) would maintain intact the mitfa and sox10 mRNAs retrieved in the single cell RNA analysis? This is why we need absolutely perfect confocal images of tumor sections with clearly visible macrophages and tumor cells to support the hypothesis that macrophages preferentially phagocyte TGFbeta-expressing melanoma cells. In addition, this claim should be supported by some in vivo live observation to test the ability of macrophages to recognize TGFbeta-expressing melanoma cells versus other melanoma cells.

The doubt that this reporter labels a subpopulation of macrophages or macrophages in a particular status of activation induced by the presence of the tumor, rather than macrophages that have phagocytosed a rare population of melanoma cells that turn on the TGFb reporter remains.

2) To understand if the overexpression of SATB2 would activate the reporter, and generate melanomas with a larger proportion of TIE:EGFP positive cells, they produced tumors overexpressing SATB2 in the TIE:EGFP background. Then, they searched for differences in the macrophage populations. Here the authors admit that TGFbeta is known to convert macrophages from a pro-inflammatory M1-like to an anti-inflammatory M2-like state, thus implicitly suggesting that the reporter, which is not specific for melanoma but could be expressed in any cell of these transgenic fish, could be activated directly in the macrophages, with no need to justify its expression with the phagocytosis of a rare population of melanoma cells expressing it.

Related to Figure 1C, sections through the melanoma would help to distinguish whether the green fluorescence signal is part of the tumor and not just the residual brain tissue that express the reporter, and that we see clearly also in extended data Figure 1D.

One of the arrowheads in Figure 1C, in the 22 wks image, does not point to gfp signal. What is the yellow halo in the middle panel mitfa:mcherry at 20 and 22 weeks?

In Extended data 1 what is the GFP halo that we see in figures C and D that does not correspond to mitfa:mCherry? Please explain.

TIE:EGFP expressing melanoma cells downregulate interferon signaling

What are the controls here that were used for comparisons, please state in the figure legend (Suppl Figure 2B). It would be interesting to know how the expression of interferon-related genes compares to surrounding non-tumoral cells for the TIE:EGFP negative tumor cells.

In Figure 4 A the mCherry signal is too high and gives a yellow halo which is not due to co-expression with TIE:EGFP (as clearly visible in the TIE:EGFP upper panel). It is necessary to reduce the intensity of the mCherry signal until no spurious yellow is seen. The same applies to Figure 4D.

---

## [Author Response]

Essential revisions (for the authors):Although the reviewers are very positive about the manuscript, there are several areas that could be explored further. There was a consensus that the role of macrophages in particular needs more data for the model to be supported.1. There are multiple issues with Figures 1 and Extended Data Figure 1 as seen in the reviewers' comments below. These relate to quality and quantification and should be addressed.2. The reviewers also suggested that Figure 3 data be improved with better confocal images and time-lapse imaging.3. In Figure 4, SATB2 data should be compared to Figures 1-3 without SATB2. Does SATB2 facilitate melanoma progression?Reviewer #1 (Recommendations for the authors):Data exists of spatially mapped transcriptomes in zebrafish melanoma:Hunter, et al. Spatially resolved transcriptomics reveals the architecture of the tumor-microenvironment interface. Nat Commun 12, 6278 (2021).Are the results in this study showing regions of active TGFb signaling in advanced melanoma consistent with existing data of endogenous expression patterns?

This is an excellent connection that would help to confirm the existence of regional TGFb signaling hubs through transcriptional analyses. In collaboration with Dr. Richard White’s lab, we analyzed this spatially resolved transcriptomics data set for regional expression of a TGFb pathway target gene signature.^1^ In order to confidently determine cells in which the TGFb pathway was activated, this required a given “gene expression spot” (10X Visium Spatial Gene Expression) that expressed many TGFb target genes. However, Dr. White’s data only identified spots in which two TGFb target genes were simultaneously expressed, lacking the resolution needed to identify regions of active TGFb signaling. This analysis was not performed targeting TGFb target genes with probes, and so the limited identification of simultaneous TGFb target gene expression in a single cell was likely due to low detection sensitivity.

The proposed model of preferential phagocytosis of TIE:GFP+ cells vs negative cells is interesting, however, the data is still somewhat preliminary. Higher resolution imaging data and possibly time-lapse would better support the model. Could the authors provide some candidate factors produced by TIE:GFP+ cells that could trigger this response?

We thank the reviewer for this feedback and agree that additional imaging would provide more solid evidence of phagocytosis. To address this, we have performed additional high-resolution confocal imaging of *TIE:EGFP*^+^ cells and macrophages in the tumor. These images reveal a subset of macrophages that express the *TIE:EGFP* reporter endogenously as well as a subset of macrophages that contain *TIE:EGFP^+^* fragments, often in endosome-like structures Figure 3—figure supplement 1 (previously Figure S4). We have found active phagocytosis to be difficult to capture in snapshots, making it challenging to quantify preferential phagocytosis by imaging. Our FACS experiment presented in Figure 3D provides a robust measurement of preferential phagocytosis of *TIE:EGFP*^+^ melanoma cells (vs. *TIE:EGFP*^-^ melanoma cells) by macrophages.

We have looked into identifying several candidate factors regulated in *TIE:EGFP^+^* cells that could trigger macrophage-mediated phagocytosis. Our single cell RNA-seq data showed differential upregulation of *serpine1* mRNA expression in *TIE:EGFP^+^* melanoma cells compared to *TIE:EGFP^-^* melanoma cells, and the *serpine1* receptor, *lrp1ab*, is upregulated in *TIE:EGFP*^+^ macrophages compared to *TIE:EGFP^-^* macrophages. This may indicate that macrophages are recruited to and subsequently phagocytose *serpine1*-expressing *TIE:EGFP^+^* melanoma cells. Indeed, SERPINE1, which encodes plasminogen activator inhibitor-1 (PAI-1), has been shown to promote cancer cell invasiveness and macrophage recruitment in an esophageal squamous cell carcinoma model.^2^ Multiple reports indicate a role for PAI-1 in macrophages recruitment and polarization.^3,4,5^ Furthermore, *TIE:EGFP^high^* macrophages differentially express *mmp14b* and the cysteine protease *legumain*, both of which have been described to promote TGFb bioavailability.^6,7,8^ These preliminary transcriptional data may indicate a mechanism by which macrophages are recruited to *TIE:EGFP^+^* regions by PAI-1 gradients. Recruited macrophages may subsequently amplify the pool of active, bioavailable TGFb through the activity of enzymes such as *mmp14b* and *lgmn*. Such a model would explain the local clustering of macrophages in TGFb-responding regions of the tumor, and their subsequent endogenous activation of TGFb signaling and phagocytic behavior in these regions. (Figure 3B and C). In the paper, we have discussed this hypothesis in the Discussion, lines 350-363.

Chronic exposure to TGFb may affect macrophage polarization, resulting in the induction of M2-like markers, including *mrc1b* and *vegfaa.* Notably, in addition to M2-like markers, *TIE:EGFP^high^* macrophages differentially express a number of genes involved in cholesterol efflux, lipid metabolism, and apoptosis including *abca1a, abca1b, lipf, npc2, adipor2, caspa,* and *casp3b*. Recent reports indicate that TGFb signaling can drive lipid droplet formation and ABCA1 expression in macrophages.^9,10^ Thus, chronic zonal TGFb exposure and uptake may skew macrophages toward a stressed, anti-inflammatory phenotype. In this model, macrophages are continuously recruited to TGFb-high zones, where they facilitate TGFb amplification, phagocytose TGFb-responsive cells, and are gradually rendered incapable of tumoricidal activity. This effect on macrophages is discussed in the Discussion section, lines 364-376.

In the Extended Data Figure 1C-D experiments, if the mitf:mCherry is transiently injected, then the conclusions may be overstated. Line 90: "TIE:EGFP was also never expressed in mitfa:mCherry high early melanomas (Extended Data Figure 1D)". How many melanomas were observed? What level of GFP expression is required to score as positive in TIE:GFP fish? The conclusions should be reconsidered because transient F0 transgenics are inherently mosaic.

It is true that *mitfa:mCherry* is transiently expressed. We have changed the word “never” to “has yet to be observed” to address this overstatement in line 92 (previously line 90). 56 tumors were observed, as stated in line 101. While FACS of each tumor may be able to give a quantitative threshold of EGFP expression by which to score *TIE:EGFP^+^* tumors, we used stereomicroscope imaging to identify *TIE:EGFP^+^* tumors due to feasibility and to track changes during further tumor progression. Using imaging, it is not easy to give a quantitative threshold of EGFP expression due to differences of EGFP brightness resulting from several factors including the depth at which the cells are located in the tumor volume. To identify *TIE:EGFP^+^* tumors, we looked for tumors in which there were several visible *EGFP^+^* cells or an *EGFP^+^* patch of cells. We also tracked the expansion of these *EGFP^+^* zones over several months, confirming that they were not imaging artifacts.

Extended data 1B images are small and difficult to evaluate. Need better images that are quantified for fluorescence intensity.

For previous Extended Data 1B (current Figure 1—figure supplement 1B) images depicting EGFP signal, we have clarified using white arrows which regions containing EGFP fluorescence are referenced, making it easier to evaluate. In addition, we have added in situ hybridization (ISH) data probing for GFP, which has been quantified for staining intensity. These changes are now reflected in Figure 1—figure supplement 1B and C.

In the Figure 1C (20 and 22 wpf) panels, why is there a yellow signal in the mitfa:mCherry channel alone in the majority of the tumor but not in the merged image?

In the *mitfa:mCherry* channel, yellow signal is due to saturated high mCherry signal. In our melanoma model, high *mitfa* expression is characteristic of zebrafish tumors. Using a standard imaging protocol (with defined zoom and exposure) to define melanoma lesions, oversaturation of *mitfa:mCherry* is displayed by the imaging software in yellow. Using Image J, we have removed the yellow oversaturation signal in these images by rendering each mCherry channel image as an 8-bit image and re-coloring the pixels in red. This way, all pixels are treated the same and the pixel intensity is retained and represented by red color intensity in the new images. We then re-merged the EGFP and new mCherry channels for all tumors in all figures.

Which data does S1B refer to in this line?Line 115: melanomas were processed for single-cell RNA-sequencing at 23 and 42 weeks post-fertilization (wpf), respectively (Figure 1C and S1B).

The two tumors that were processed for single-cell RNA-sequencing and referred to in previous Line 115 are (1) the tumor shown in manuscript Figure 1C (dissected at 23 wpf) and (2) the tumor shown in manuscript Figure 1—figure supplement 2B (dissected at 42 wpf).

Could the authors describe the marco marker and its significance here?Line: 120 melanoma cells expressing mitfa and/or sox10, but we identified a marco-expressing TIE:EGFP+.

*Marco* is a known marker of macrophages, as well as a marker of M2-polarized macrophages. To be more straightforward in identifying the macrophage population in Figures 3 and 4 (and previous line 120), we defined the macrophage cluster using *mpeg1.1* expression instead of *marco* expression.

Does the analysis of Figure S2B include macrophages or is it only the melanomas? Is there any independent validation of the GSEA results? This is regarding:Line: 130 in TIE:EGFPhigh cells were interferon α and γ (Figure S2B).

Manuscript Figure 2—figure supplement 1B (previously Figure S2B) is labeled “Melanoma Cells,” it does not include macrophages. To provide an independent validation of the GSEA results showing down-regulation of IFNa and IFNγ pathway activation in *TIE:EGFP^high^* melanoma cells, we have isolated *TIE:EGFP^high^* and *TIE:EGFP^-^* melanoma cells from two additional tumors by FACS and performed qPCR for several IFN-response genes. This data confirms that IFN target genes go down in *TIE:EGFP^high^* tumor cells.

**Author response image 1. sa2fig1:** qPCR shows that IFN target genes are down-regulated in *TIE:EGFP^high^* melanoma cells compared to *TIE:EGFP^-^* melanoma cells. Two *Tg(mitfa:BRAF^V600E^);p53*^-*/*-^*;mitfa*^-*/*-^ tumors expressing *TIE:EGFP*, *mitfa:BFP* and *mpeg:mCherry* were dissociated for FACS. We sorted two populations of cells: *TIE:EGFP^high^ mitfa:BFP^+^ mpeg:mCherry^-^,* and *TIE:EGFP^-^ mitfa:BFP^+^ mpeg:mCherry^-^.* Here, we gated for *mitfa:BFP^+^* melanoma cells, gating out the macrophage population which may contain *mitfa:BFP* due to tumor cell phagocytosis. We isolated RNA from these sorted populations, performed cDNA synthesis, and performed qPCR. We used housekeeping gene *b-actin* to normalize values according to loading differences. Here, we confirmed that 4 major IFN genes discussed in the manuscript are down-regulated*: irf7, irf1b, stat1b, and b2m.* We included 3 additional IFN genes here to further validate the IFN pathway trend: *mxa, mxb, and isg15.* Fold change of *mitfa* is included here as a negative control, as we do not expect its expression to change between the two conditions. N=2 tumors, 3 technical replicates each. 2-tailed, unpaired t-tests were performed.

In the following line, which genes are referred to specifically? Are these genes associated with a specific table or figure?Line: 293 of melanoma patients exhibit up-regulation of these genes52,53.

This sentence refers to the 29 upregulated chronic TGFb genes shown in manuscript Figure 2C (first 29 genes listed on the dot plot).

Are there experiments that could better support the following statement:Line: 306 Our work identified macrophage subclusters that contain low levels of mitfa and/or sox10 expression suggestive of phagocytosis of melanoma cells.

To further support the claim that *mitfa* and *sox10* are expressed in macrophages due to melanoma cell phagocytosis, we have performed additional high-resolution imaging of *TIE:EGFP*^+^ cells and macrophages in zebrafish tumors (Figure 3—figure supplement 1). This imaging reveals fragments of *TIE:EGFP* within macrophages, occasionally contained in endosome-like structures. It also reveals co-localization of *mitfa:BFP, mpeg:mCherry,* and *TIE:EGFP*, suggesting phagocytosis of TGFb responsive melanoma cells.

Additionally, we isolated *TIE:EGFP^+^* and *TIE:EGFP*^-^ macrophages (using the *mpeg:mCherry* reporter) by FACS and performed qPCR for *mitfa* and *sox10.* This data (shown in manuscript Figure 3—figure supplement 2) shows that *mitfa* and *sox10* expression is enriched in *TIE:EGFP^+^* macrophages compared to *TIE:EGFP*^-^ macrophages, further supporting that macrophages preferentially phagocytose *TIE:EGFP*^+^ melanoma cells.

Is there a GFP/mCherry merged image to support this statement, as it is not clear from the figure?311 Figure 3C, there is co-localization of mpeg:mCherry and TIE:EGFP signal.

Thank you for requesting this clarification. To address this, we have included a merged image of the EGFP and mCherry channels in manuscript Figure 3C and have indicated the region of co-localization of *mpeg:mCherry* and *TIE:EGFP* with a white arrow.

The following figure legend is insufficient and needs more details (e.g. live or fixed sample? Optical section or z projection? Single frame of time-lapse?):Line: 943 Supplement Figure 4. Example of a macrophage actively phagocytosing a TIE:EGFP+ cell in a melanoma.

To address this, we have added additional imaging information to the legend for manuscript Figure 3—figure supplement 1 (previously S4), such as methods for live imaging of zebrafish tumors, and optical sectioning vs. Z projection.

Reviewer #2 (Recommendations for the authors):1. Figure 1, Extended data 1 and supplemental Figure 1: better characterization of the reporter lines is important at different ages and in melanomas. Only one melanoma is shown in Figure 1C and another one in S1B, and, curiously, the pattern of TIE:EGFP expression is completely different: clusters vs homogenous expression. More pictures and quantitation of TIE:EGFP+ and TIE:EGFP- cells are required.

To show more characterization of the *TIE:EGFP* reporter in melanoma, we have included an additional supplemental figure (Figure 1—figure supplement 3) showing its expression in 5 additional melanomas. Almost all tumors have only a small cluster of cells in which the *TIE:EGFP* reporter is activated, however an occasional tumor activates the reporter as highly as that represented in manuscript Figure 1—figure supplement 2B (previously Figure S1B). Although imaging was ideal for our purposes of tracking changes of EGFP expression over time during tumor progression, FACS would be able to more accurately quantify *TIE:EGFP^+^* and *TIE:EGFP^-^* cells in the tumor, since imaging does not capture internal EGFP expression. As written in previous line 192 (current line 202), FACS determined that less than 1% of total cells from a tumor are EGFP^+^ (n=3).

2. Figure 3 and results about macrophages: Figure 3A: most macrophages do not express mitfa and sox10. Figure 3B is not convincing: I do not see any white cells that would represent macrophages taken up TIE:EGFP+ melanoma cells. I can see several green macrophages that may indicate that macrophages express the TIE:EGFP construct. Additional pictures and video, if possible are needed.

We thank the reviewers for this comment and agree this should be clarified. We do not expect all/most macrophages to express *mitfa* and *sox10*. Because only a subset of macrophages expresses these genes, this supports the hypothesis that some of them have phagocytosed melanoma cells, rather than the possibility that all macrophages express these genes endogenously. We have clarified in the legend for manuscript Figure 3B that if a macrophage expresses the *TIE:EGFP* reporter endogenously, we would expect to see the entire volume of the macrophage to express EGFP, rather than fragments. In lines 183-198, we have clarified that the majority of *TIE:EGFP^+^* macrophages contain EGFP^+^ fragments, suggesting that they phagocytosed *TIE:EGFP^+^* cells.

In three out of thirteen tumors imaged by confocal microscopy, we did capture a cluster of *TIE:EGFP^+^ mitfa:BFP^+^* melanoma cells closely co-localized with macrophages, indicating that macrophages are engulfing these cells (Figure 3B, Figure 3—figure supplement 1B). To clarify which macrophages have engulfed *TIE:EGFP^+^* melanoma cells in manuscript Figure 3B and Figure 3—figure supplement 1, we have added boxes around areas including white cells (which represent *mitfa:BFP^+^ TIE:EGFP^+^ mpeg:mCherry^+^* cells). It is rare to capture active phagocytosis in snapshots (as seen in Figure 3—figure supplement 1C). However, the FACS experiment shown in manuscript Figure 3D and qPCR in manuscript Figure 3—figure supplement 2 confirms that macrophages preferentially phagocytose *TIE:EGFP^+^* melanoma cells over *TIE:EGFP^-^* melanoma cells.

In the majority of tumors, however, most *TIE:EGFP*^+^ fragments within macrophages appear *mitfa:BFP^-^* via imaging. We have mentioned in the manuscript, particularly in lines 183-198 and in the Figure 3 legend, that macrophages likely phagocytose *TIE:EGFP*^+^ non-melanoma cells (in addition to *TIE:EGFP*^+^ melanoma cells). These non-melanoma cells may not have been identified in our SORT-seq data because the tissue dissociation protocol used is ideal for tumor cell viability, but not for that of various immune cell populations. Indeed, by confocal imaging, the majority of *TIE:EGFP^+^* signal appears to be within clusters of macrophages, while FACS analysis and SORT-seq analysis shows that a large proportion of *TIE:EGFP^+^* cells are tumor cells. This may be attributed to the technicality that tumor dissociation using Liberase (used for FACS and SORT-seq) favors tumor cell survival/dissociation over immune cell populations.

One alternative reason we may not often observe (via imaging) three-way co-localization between *TIE:EGFP, mpeg:mCherry,* and *mitfa:BFP* may be attributed to timing of observation. We most often observe macrophages with *TIE:EGFP* fragments, which may be the remnants of previously engulfed *TIE:EGFP^+^;mitfa:BFP^+^* tumor cells, and the BFP expression is no longer strong enough to see at the time of imaging. *Mitfa:BFP* is expressed mosaically in these tumor cells and may not be expressed in every tumor cell. Because we do occasionally observe macrophages engulfing clusters of *TIE:EGFP^+^* melanoma cells, and that FACS confirms that macrophages contain *TIE:EGFP^+^ mitfa:BFP^+^* fragments, we are confident that melanoma cells are one *TIE:EGFP^+^* population being phagocytosed by macrophages. We focused our studies on this particular interaction to show that macrophages preferentially phagocytose *TIE:EGFP*^+^ melanoma cells compared to *TIE:EGFP*^-^ melanoma cells.

3. Figure 3D: I do not understand why Q1 represent non-melanoma cells that were phagocytosed. I think these are non-phagocytosing macrophages. So, Q2 has to be plotted not Q1+Q2. Anyway, I think that allotransplant experiments of melanoma cells from TIE:EGFP fish need to be performed in mpeg1:cherry recipient, and vice versa, to demonstrate that macrophages are phagocytosing TIE:GFP+ cells rather than expressing the construct. In addition.

We appreciate this being brought to our attention, and we agree that this alternative hypothesis is likely. Although imaging shows that the majority of *TIE:EGFP^+^* macrophages contain only fragments of EGFP, suggesting phagocytosis, it is possible that Q1 can represent a subset of non-phagocytosing macrophages that express the *TIE:EGFP* construct endogenously, or macrophages that phagocytosed *TIE:EGFP^+^* non-melanoma cells. We have clarified this in the figure legend and re-plotted Q2 only in manuscript Figure 3D.

While an allotransplant would help answer this question of phagocytosis, this experiment would have caveats in our system. Because we do not have isogenic lines expressing these reporters, macrophages from the recipient fish would phagocytose transplanted *TIE:EGFP^+^* cells due to foreign recognition in a “graft vs. host” response, confounding our results.

4. Figure 4 and SATB2 data: these data should be compared with those obtained in Figure 1-3 w/o SATB2. Does SATB2 facilitate melanoma progression? Is there any difference in the TGFB targets between control and SATB2 TIE:EGFP+ cells?

*SATB2* overexpression in our zebrafish model does facilitate melanoma progression, as stated in previous lines 204-206 (currently line 215-217), in which we referenced a paper previously published in our lab showing this data. We re-worded this sentence to include that *SATB2* overexpression accelerates melanoma onset. We have also revised manuscript Figure 4 to include images of tumor development and *TIE:EGFP* activation in a control *MCR:MCS* tumor for reference.

There are differences in TGFb target genes in *TIE:EGFP^+^* tumor cells between *MCR:MCS* and *MCR:SATB2* tumors, although some genes are shared. Shown below is a table listing target genes that are upregulated in *TIE:EGFP^+^* melanoma cells in a control *MCR:MCS* tumor and a *MCR:SATB2* tumor, including those that are shared between the two genotypes (Author response table 1). In manuscript Figure 4C, we have indicated the shared genes in red text.

**Author response table 1. sa2table1:** List of top up-regulated genes in *TIE:EGFP^+^* melanoma cells of *MCR:MCS* and *MCR:SATB2* tumors, as defined by SORT-seq. These genes are also listed in manuscript Figures 2C and 4C.

MCS	Shared	SATB2
arhgdia	bsg	mmp14b
lama4	col6a1	igf2r
lrp1ab	itgb2	igfbp3
rhoq	marcksl1a	igfbp5b
rnd3a	nid1a	pak1
ddr1	rhoab	pxnb
fgd1	sparc	nme4
itga4	lamb1a	ephb2b
itga5	tnc	cdh2
col1a2		smad7
dv1b		sox2
cdkn2a		vim
cdkn2b		rac1a
serpine1		lrrc17
fn1a		sema3fb
snai1b		ctnnb1
twist1b		
snai2		

5. Lines 286-295: it is speculated that the novel chronic TGFB targets are involved in cell migration. I think it would be a nice complement to perform an allotransplant of TIE:EGFP+ and TIE:EGF- melanoma cells and study their aggressiveness.

We agree that an allotransplant of *TIE:EGFP^+^* melanoma cells would be a nice experiment to determine if they are more migratory than *TIE:EGFP^-^* melanoma cells. However, this is beyond the scope of this paper. In this manuscript, we merely speculate that TGFb signaling may promote migration of melanoma cells, and we plan to directly determine this in future experiments.

Reviewer #3 (Recommendations for the authors):1) The confocal pictures in Figure 3B are not convincing. We cannot see the edge of the cells to make sure that they are not debris. At the same time, how do we expect that melanoma cells that have been phagocytosed by macrophages and that are present in fragments (only puncta of TIE:EGFP signal are seen in macrophages) would maintain intact the mitfa and sox10 mRNAs retrieved in the single cell RNA analysis? This is why we need absolutely perfect confocal images of tumor sections with clearly visible macrophages and tumor cells to support the hypothesis that macrophages preferentially phagocyte TGFbeta-expressing melanoma cells. In addition, this claim should be supported by some in vivo live observation to test the ability of macrophages to recognize TGFbeta-expressing melanoma cells versus other melanoma cells.The doubt that this reporter labels a subpopulation of macrophages or macrophages in a particular status of activation induced by the presence of the tumor, rather than macrophages that have phagocytosed a rare population of melanoma cells that turn on the TGFb reporter remains.

We thank the reviewer for this comment and have made revisions to address this concern. We have performed additional confocal imaging of zebrafish tumors, for which we have included images in manuscript Figure 3—figure supplement 1. This consistently shows *TIE:EGFP^+^* fragments contained in *mpeg:mCherry^+^* macrophages, representing macrophages that have phagocytosed *TIE:EGFP^+^* cells. These EGFP^+^ fragments are not debris/imaging artifacts, since they are observed within the bounds of *mpeg:mCherry*^+^ cells, they are not fluorescent in every channel (as debris usually is), and they are contained in endosome-like structures in macrophages (manuscript Figure 3—figure supplement 1A, right). We have clarified in lines 183-198 and Figure 3 legend that macrophages likely phagocytose *TIE:EGFP*^+^ non-melanoma cells (in addition to *TIE:EGFP*^+^ melanoma cells), which may be why we observe a subset of macrophages containing *TIE:EGFP* fragments without *mitfa:BFP* fragments. One alternative reason we may not often observe (via imaging) three-way co-localization between *TIE:EGFP, mpeg:mCherry,* and *mitfa:BFP* may be attributed to timing of observation. We most often observe macrophages with *TIE:EGFP* fragments, which may be the remnants of previously engulfed *TIE:EGFP^+^;mitfa:BFP^+^* tumor cells, and the BFP expression is no longer strong enough to see at the time of imaging. *Mitfa:BFP* is expressed mosaically in these tumor cells and may not be expressed in every tumor cell.

We have included additional images of tumors in which macrophages do contain *TIE:EGFP* and *mitfa:BFP* fragments (represented by white cells contained in boxes in manuscript Figure 3B and Figure 3—figure supplement 1B). Together with the FACS experiment depicted in Figure 3D, this data confirms that macrophages phagocytose *TIE:EGFP^+^* melanoma cells. While our imaging data suggests that macrophages likely phagocytose *TIE:EGFP*^+^ non-melanoma cells as well, we decided to focus our studies on the melanoma cell population to show that macrophages preferentially phagocytose *TIE:EGFP*^+^ melanoma cells compared to *TIE:EGFP*^-^ melanoma cells. It is possible for macrophages that have recently phagocytosed *TIE:EGFP^+^* melanoma cells to maintain *mitfa* and *sox10* mRNAs, if these cells are contained in phagosomes that have not yet fused with lysosomes and undergone degradation. This is likely why *mitfa* and *sox10* mRNAs are retrieved in single cell RNA analysis in a subset of macrophages. The retainment of *mitfa* and *sox10* transcripts was validated in an additional qPCR we performed of *TIE:EGFP^+^* and *TIE:EGFP*^-^ macrophages, as described in Figure 3—figure supplement 2.

In addition to clarifying the phagocytic behavior of *TIE:EGFP^+^* cell populations, our additional imaging revealed a subset of macrophages that express *TIE:EGFP* endogenously (manuscript Figure 3—figure supplement 1A, left). A description of these two subsets of *TIE:EGFP^+^* macrophages is included in lines 183-198.

2) To understand if the overexpression of SATB2 would activate the reporter, and generate melanomas with a larger proportion of TIE:EGFP positive cells, they produced tumors overexpressing SATB2 in the TIE:EGFP background. Then, they searched for differences in the macrophage populations. Here the authors admit that TGFbeta is known to convert macrophages from a pro-inflammatory M1-like to an anti-inflammatory M2-like state, thus implicitly suggesting that the reporter, which is not specific for melanoma but could be expressed in any cell of these transgenic fish, could be activated directly in the macrophages, with no need to justify its expression with the phagocytosis of a rare population of melanoma cells expressing it.

Indeed, additional confocal imaging reveals a subset of macrophages that express *TIE:EGFP* endogenously, which populations will be interesting to focus on going forward (manuscript Figure 3—figure supplement 1A, left). This imaging also reveals a subset of macrophages that phagocytose *TIE:EGFP^+^* cells (manuscript Figure 3B and Figure 3—figure supplement B), which is a separate interesting phenomenon discussed in this manuscript. We have included a description of these two *TIE:EGFP^+^* macrophage populations in lines 183-198.

Related to Figure 1C, sections through the melanoma would help to distinguish whether the green fluorescence signal is part of the tumor and not just the residual brain tissue that express the reporter, and that we see clearly also in extended data Figure 1D.

We are confident that the patch of EGFP fluorescence indicated by an arrowhead near the brain in manuscript Figure 1C is not residual brain fluorescence because it is expressed superficially and is expressed in individual cells, rather than the typical hazy, internal brain fluorescence pattern we see. We also observed this EGFP^+^ patch grow over the course of a few weeks. However, to further show that tumors do indeed contain clusters of *TIE:EGFP* activation, we have included an additional supplemental figure (Figure 1—figure supplement 3) displaying images of five additional advanced tumors, including those at different anatomical locations.

One of the arrowheads in Figure 1C, in the 22 wks image, does not point to gfp signal. What is the yellow halo in the middle panel mitfa:mcherry at 20 and 22 weeks?

This arrowhead in manuscript Figure 1C does point to EGFP^+^ cells, but they are hard to see at this magnification, so we deleted the arrowhead to minimize confusion.

In the *mitfa:mCherry* channel, yellow signal is due to saturated high mCherry signal. In our melanoma model, high *mitfa* expression is characteristic of zebrafish tumors. Using a standard imaging protocol (with defined zoom and exposure) to define melanoma lesions, oversaturation of *mitfa:mCherry* is displayed by the imaging software in yellow. Using Image J, we have removed the yellow oversaturation signal in these images by rendering each mCherry channel image as an 8-bit image and re-coloring the pixels in red. This way, all pixels are treated the same and the pixel intensity is retained and represented by red color intensity in the new images. We then re-merged the EGFP and new mCherry channels for all tumors in all figures.

In Extended data 1 what is the GFP halo that we see in figures C and D that does not correspond to mitfa:mCherry? Please explain.

The EGFP halo in previous Extended Data 1 (current Figure 1—figure supplement 1) indicates endogenous *TIE:EGFP* reporter expression in the spine (Figure 1—figure supplement 1D) and brain (Figure 1—figure supplement 1E). We have now clarified this in the legend for this figure.

TIE:EGFP expressing melanoma cells downregulate interferon signalingWhat are the controls here that were used for comparisons, please state in the figure legend (Suppl Figure 2B). It would be interesting to know how the expression of interferon-related genes compares to surrounding non-tumoral cells for the TIE:EGFP negative tumor cells.

The internal control for this experiment was *TIE:EGFP*^-^ melanoma cells, to which we compared interferon stimulated gene (ISG) expression in *TIE:EGFP^low^* and *TIE:EGFP^high^* melanoma cells from the same tumor. We have clarified this in the legend of manuscript Figure 2—figure supplement 1 (previously S2B).

Using a scRNA-seq dataset we had available in the lab that contained melanocytes from zebrafish normal skin, early melanoma, and an advanced tumor, we analyzed relative interferon-related genes at these different stages (Author response image 2). Generally, it appears that interferon-related genes often increase in tumors compared to normal skin and precursor lesions. While IFN genes are generally up-regulated in melanoma cells within a tumor, the data in this manuscript describes a rare subset of tumor cells that up-regulate TGFb and thus down-regulate IFN.

**Author response image 2. sa2fig2:** ScRNA-seq shows that expression of IFN target genes increases in tumors compared to normal skin and precursor lesions. *Mitfa:BFP^+^* melanocytes were sorted via FACS from normal skin, an early melanoma, and an advanced melanoma and 10X single-cell RNA-seq was performed. Data was analyzed using Seurat and cell types were called based on gene expression signatures. Depicted here are violin plots of several IFN target genes in melanocytes across the three stages.

In Figure 4 A the mCherry signal is too high and gives a yellow halo which is not due to co-expression with TIE:EGFP (as clearly visible in the TIE:EGFP upper panel). It is necessary to reduce the intensity of the mCherry signal until no spurious yellow is seen. The same applies to Figure 4D.

In manuscript Figure 4A, the yellow signal in the *mitfa:mCherry* channel represents saturated mCherry signal, indicative of characteristic *mitfa* high state of our zebrafish tumors. Using Image J, we have removed the yellow oversaturation signal in these images by rendering each mCherry channel image as an 8-bit image and re-coloring the pixels in red. This way, all pixels are treated the same and the pixel intensity is retained and represented by red color intensity in the new images. We have excluded the merged images in this panel to minimize confusion about the yellow signal, and to make space for a panel including development of a reference *MCR:MCS* tumor.

References:

Hunter, M.V., Moncada, R., Weiss, J.M. et al. Spatially resolved transcriptomics reveals the architecture of the tumor-microenvironment interface. *Nat Commun* 12, 6278 (2021).

Sakamoto, H., Koma, Yi., Higashino, N. et al. PAI-1 derived from cancer-associated fibroblasts in esophageal squamous cell carcinoma promotes the invasion of cancer cells and the migration of macrophages. Lab Invest 101, 353–368 (2021).

Kubala, M. H., Punj, V., Placencio-Hickok, V. R., Fang, H., Fernandez, G. E., Sposto, R., & DeClerck, Y. A. Plasminogen Activator Inhibitor-1 Promotes the Recruitment and Polarization of Macrophages in Cancer. Cell reports, 25(8), 2177–2191 (2018).

Honjo, K., Munakata, S., Tashiro, Y., Salama, Y., Shimazu, H., Eiamboonsert, S., Dhahri, D., Ichimura, A., Dan, T., Miyata, T., Takeda, K., Sakamoto, K., Hattori, K., & Heissig, B. Plasminogen activator inhibitor-1 regulates macrophage-dependent postoperative adhesion by enhancing EGF-HER1 signaling in mice. FASEB journal : official publication of the Federation of American Societies for Experimental Biology 31(6), 2625–2637 (2017).

Baumeier, C., Escher, F., Aleshcheva, G. et al. Plasminogen activator inhibitor-1 reduces cardiac fibrosis and promotes M2 macrophage polarization in inflammatory cardiomyopathy. Basic Res Cardiol 116, 1 (2021).

Bai, P. et al. Macrophage-Derived Legumain Promotes Pulmonary Hypertension by Activating the MMP (Matrix Metalloproteinase)-2/TGF (Transforming Growth Factor)-β1 Signaling. Arteriosclerosis, Thrombosis, and Vascular Biology. 39 (4) (2019).

Sounni, N.E., Dehne, K., van Kempen, L., Egeblad, M., Affara, N.I., Cuevas, I., Wiesen, J., Junankar, S., Korets, L., Lee, J., Shen, J., Morrison, C.J., Overall, C.M., Krane, S.M., Werb, Z., Boudreau, N., Coussens, L.M. Stromal regulation of vessel stability by MMP14 and TGFbeta. Dis Model Mech. 3, (2010).

Robertson, I.B., Rifkin, DB. Regulation of the Bioavailability of TGF-β and TGF-β-Related Proteins. Cold Spring Harb Perspect Biol. 8, (2016).

Hu, Y. W., Wang, Q., Ma, X., Li, X. X., Liu, X. H., Xiao, J., Liao, D. F., Xiang, J., & Tang, C. K. TGF-beta1 up-regulates expression of ABCA1, ABCG1 and SR-BI through liver X receptor α signaling pathway in THP-1 macrophage-derived foam cells. Journal of atherosclerosis and thrombosis, 17(5), 493–502 (2010).

Bose, D., Banerjee, S., Chatterjee, N., Das, S., Saha, M., & Saha, K. D. Inhibition of TGF-β induced lipid droplets switches M2 macrophages to M1 phenotype. Toxicology in vitro : an international journal published in association with BIBRA, 58, 207–214 (2019).